# VINCIE: Unlocking In-context Image Editing from Video

**Leigang Qu**[1]   **Feng Cheng**[2]   **Ziyan Yang**[2]   **Qi Zhao**[2]   **Shanchuan Lin**[2]
**Yichun Shi**[2]   **Yicong Li**[1]   **Wenjie Wang**[1]   **Tat-Seng Chua**[1]   **Lu Jiang**[2]
[1]National University of Singapore     [2]ByteDance Seed
https://vincie2025.github.io/

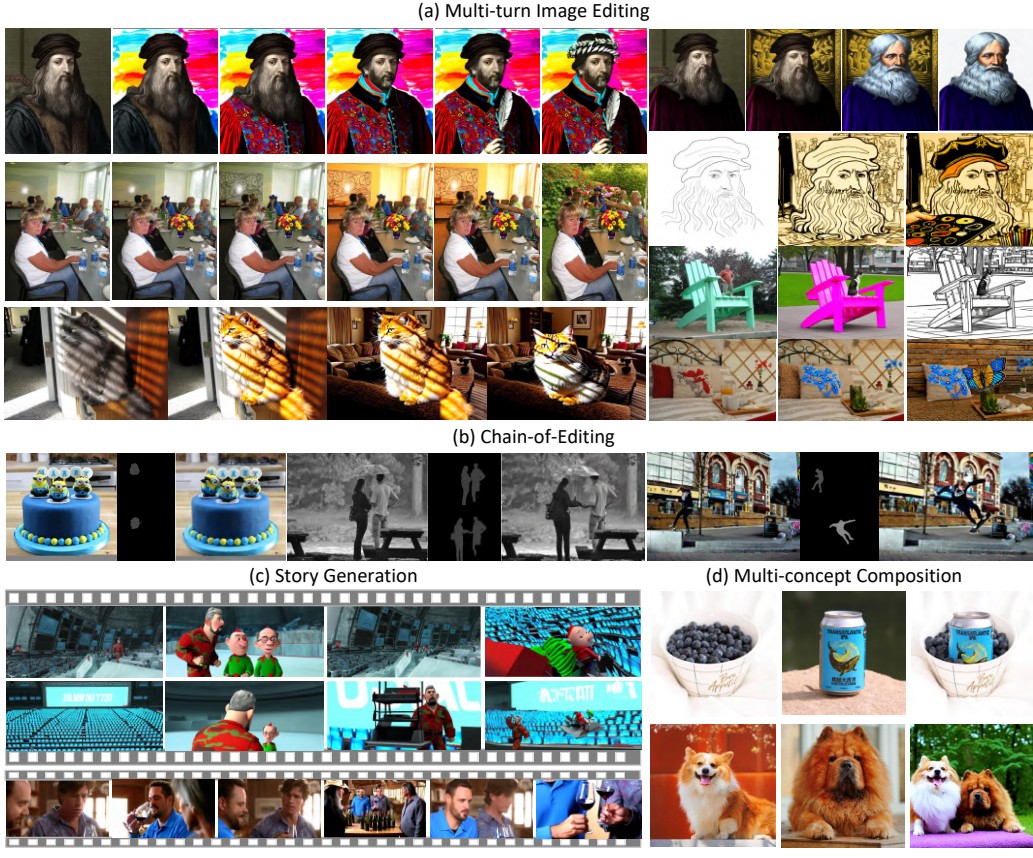

Figure 1: By learning from videos, our method could attain universal in-context editing and generation abilities to handle various practical creation scenarios.

## Abstract

In-context image editing aims to modify images based on a contextual sequence comprising texts and images. Existing methods typically depend on task-specific pipelines and expert models (*e.g.*, segmentation and inpainting) to curate training data. In this work, we explore whether an in-context image editing model can be learned directly from videos. Toward this end, we introduce a scalable approach to annotate videos as interleaved multimodal sequences. To effectively learn from this data, we design three proxy tasks: next-image prediction, current segmentation prediction, and next-segmentation prediction. Additionally, we propose a novel multi-turn image editing benchmark to advance research in this area. Extensive experiments demonstrate that our model exhibits strong in-context image editing capabilities and achieves state-of-the-art results on two multi-turn image editing benchmarks. Despite being trained exclusively on videos, our model also shows

promising abilities in multi-concept composition, story generation, and chain-of-editing applications.

# 1 INTRODUCTION

Recent research has devoted significant effort to image editing, which enables users to generate images that closely follow editing instructions provided in text prompts. The performance of image editing models largely depends on the high-quality training data, typically composed of three elements: an input image, a text prompt describing the desired modification, and the corresponding edited image (Brooks et al., 2023; Shi et al., 2024a; Xiao et al., 2024; Wei et al., 2024; Han et al., 2024b; Xia et al., 2024; Liu et al., 2025). To collect such paired image data at scale, various methods have been proposed, including generating image grids (Wu et al., 2025c), leveraging diffusion denoising processes (Brooks et al., 2023), and developing specialized models or tools to extract before-and-after image pairs from the web (Hertz et al., 2022; Zhuang et al., 2024; Boesel & Rombach, 2024).

Very recently, the problem of *in-context image editing* (OpenAI, 2025b) has garnered growing interest in the research community. In this setting, a target image is generated based on a contextual sequence of text prompts and previously generated images. Unlike single-turn image editing, in-context image editing supports multi-turn interactions, enabling users to iteratively refine images while maintaining visual consistency throughout the editing process. A key challenge lies in acquiring contextualized training data that includes coherent sequences of text and images, Existing approaches to mine single-turn image editing (Brooks et al., 2023; Wu et al., 2025c; Hertz et al., 2022; Zhuang et al., 2024; Boesel & Rombach, 2024) struggle to construct meaningful long-form content that is capable of capturing the dependencies and evolving intent that emerge over multiple editing steps. The lack of contextualized, quality training data remains a significant barrier to progress in this area of research.

In this paper, we approach in-context image editing from a different perspective and investigate the following research question: *Can a meaningful in-context image editing model be learned solely from videos, without using any standalone images?* Our intuition is that videos, as a rich source of multimodal information, inherently contain a long duration of visual dynamics that might facilitate the learning of multi-turn interactions. For instance, changes within a scene, such as objects entering or exiting the frame, shifts in camera focus, or character actions, provide implicit cues for learning operations like addition, removal, and modification in image editing.

To this end, we propose an approach that natively learns transitions from video data, named Video-driven IN-Context Image Editing (**VINCIE**). Unlike conventional image editing methods that rely on separately collected pairs of pre- and post-editing images for training, we choose not to alter the video, *i.e.*, we train on native video data (only natural videos as the source of visual modality), but instead provide the model with detailed annotations that describe the transitions or actions occurring within the scene. Since our method eliminates the need for paired data collection and relies solely on video, it can be trivially scaled using the vast amount of video data readily available on the web.

Specifically, we first sample a few coherent frames from a video scene, annotate the visual transitions, and identify Regions of Interest for editing (RoEs) using a pretrained Vision-Language Model (VLM). Additionally, we employ Grounding-DINO (Liu et al., 2024b) and SAM2 (Ravi et al., 2024) to generate RoE segmentation masks based on textual descriptions of the transitions. This process establishes our training samples, which capture context and form an interleaved multimodal sequence. Next, we train a Diffusion Transformer (Peebles & Xie, 2023) with full attention as our primary implementation and additionally design a variant with block-wise causal attention, which applies bidirectional attention within each modality (frame, text, and segmentation mask) and causal attention across modalities. Both variants are compared to provide a direct assessment of their differences.

Finally, to enhance the model's learning of contextual dependencies, we design three proxy tasks: (1) next-image prediction, which serves as the primary task in training; (2) current segmentation prediction, which enables the model to understand which regions have changed; and (3) next segmentation prediction, which prepares the model to anticipate where changes are likely to occur.

Extensive experiments show that our model, trained solely on video data, demonstrates strong in-context image editing capabilities and outperforms existing baselines on the multi-turn image editing tasks. Scaling up the model and training data leads to substantial performance gains—for example, the success rate at the challenging 5-turn editing increases from 5% to 22% when scaling the training

data from 0.25M to 10M sessions—demonstrating the scalability of our approach enabled by native video data. Notably, to the best of our knowledge, this is the first work to demonstrate the feasibility of learning an in-context image editing model solely from video data, while also showcasing the scalability benefits of this approach.

We find that our model can learn disentangled representations of visual changes (*e.g.*, object appearance/disappearance, posture shifts, and orientation changes) purely from patterns inherent in video data. It also demonstrates reasonable generalization to scenarios that are less common in natural video, such as background changes, attribute modifications, and multi-concept compositions. As an additional benefit, our model can be used for generating consistent frames for storytelling through in-context editing.

## 2 RELATED WORK

**Data Construction Methods for Image Editing.** Constructing image editing datasets requires first designing clear and diverse editing instructions that articulate the intended visual modifications. Based on these instructions, paired image examples are then created, consisting of original images and their corresponding edited versions that reflect the specified transformations. Single-turn image editing methods (Hertz et al., 2022; Brooks et al., 2023; Sheynin et al., 2024; Shi et al., 2024a; Zhao et al., 2024; Wei et al., 2024; Hui et al., 2024; Yang et al., 2024b; Jin et al., 2024) use pre-trained off-the-shelf models (Ramesh et al., 2022; Rombach et al., 2022; Brown et al., 2020; Sauer et al., 2024) to construct paired data for image editing. For example, InstructPix2Pix (Brooks et al., 2023) leverages GPT-3 (Brown et al., 2020) for generating editing instructions and Stable Diffusion v1.5 (Rombach et al., 2022) for paired image data generation. UltraEdit creates editing instructions using LLMs and combines grounding models (Kirillov et al., 2023; Liu et al., 2024b) with SDXL-Turbo (Sauer et al., 2024) to produce region-based editing samples. Our approach relies on learning transitions from videos without manually crafted paired data pipelines, bringing impressive scalability.

**Learning from Video for Image Generation.** Video Frames naturally exhibit consistency across characters, objects, and scenes, which has inspired recent efforts to construct source and target images from sampled video frames. Leveraging such frame-based data has proven beneficial for enhancing consistency in image generation tasks, such as instructive image editing (Chen et al., 2024d; Krojer et al., 2024), interactive image editing (Zhang et al., 2025a; Shi et al., 2024b), streamlining image editing (Alzayer et al., 2024), and object-level image customization (Chen et al., 2024c). The most recent work, *e.g.*, RealGeneral (Lin et al., 2025) and UES (Chen et al., 2024a), explored the temporal in-context consistency within video foundation models (Yang et al., 2024c) for universal image generation and editing. Despite notable progress, existing methods typically rely on only two frames per video, overlooking richer, long-range contextual information. Furthermore, they often depend on task-specific data construction pipelines (Chen et al., 2024d; Zhang et al., 2025a; Chen et al., 2024c), limiting their universality and scalability. In this work, we propose constructing session-wise data with long, interleaved image-text context from native videos, and leverage it for pre-training or mid-training to learn the inherent consistency and transformations in abundant multimodal sequences.

## 3 METHODOLOGY

### 3.1 INTERLEAVED MULTIMODAL SEQUENCE CONSTRUCTION

Figure 2 shows an overview of our data construction pipeline. Starting with a video, we sparsely sample $K$ frames $(I_0, \ldots, I_K)$ and use a vision-language model (VLM) to generate textual visual transitions $T_i$ describing the change from frame $I_i$ to $I_{i+1}$. To better capture the Regions-of-interest for editing (RoEs), we additionally annotate segmentation masks $M_i$ and $M_{i+1}$, which identify the changing objects in $I_i$ and $I_{i+1}$, respectively. Combining these elements, we construct the multimodal sequence $(I_0, T_0, T_{m0}, M_{00}, T_{m1}, M_{01}, I_1, \ldots, I_K)$. $T_{m0}$ and $T_{m1}$ are predefined prompts such as "generate the mask of changing areas in the source image" and "generate the mask of changing areas in the target image".

**Frame Sampling**. We use a hybrid sampling strategy: 1) *Equal-interval sampling*, which selects frames at fixed time intervals (*e.g.*, 3 sec), and 2) *Fixed-frame sampling*, which uniformly samples

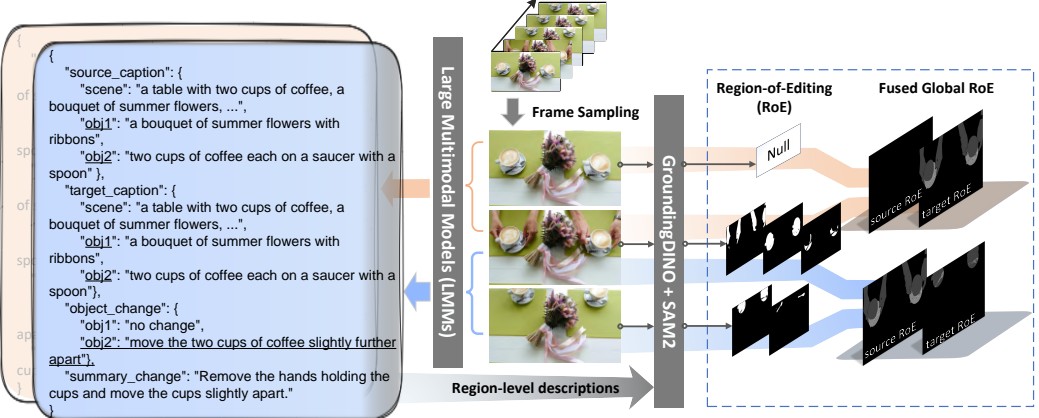

Figure 2: Our session data construction pipeline. We use a VLM to annotate the visual transitions. We then use the generated textual descriptions to prompt GroundingDINO+SAM2, extracting segmentation masks for the edited regions.

a fixed number (*e.g.*, $2 \leq n \leq 6$) of frames regardless of video duration. This approach is used to capture both subtle object-level changes and significant scene-level transitions.

**Visual Transition Annotation**. To describe visual transitions between frames, we use chain-of-thought (CoT) prompting (Wei et al., 2022) to instruct a VLM to perform visual transition annotation: 1) generate detailed and coherent descriptions of each frame from multiple aspects (*e.g.*, characters, objects, attributes, interactions, scenes, and environments); 2) identify semantic and visual differences between the two frames from the above aspects; 3) and summarize all the differences into a concise, instruction-style statement $T_i$ suitable for guiding editing. Unlike existing interleaved datasets (Zhu et al., 2023; Laurençon et al., 2023; Chen et al., 2024b) derived from web documents and retrieval tools, our dataset is built from native videos, ensuring stronger textual and visual coherence.

**Segmentation Annotation and Encoding**. We explicitly annotate Regions-of-Editing (RoEs) in both adjacent frames $I_i$ and $I_{i+1}$. Specifically, we leverage region-level descriptions (*i.e.*, characters and objects) in the visual transition annotation as input to GroundingDINO (Liu et al., 2024b) and SAM 2 (Ravi et al., 2024) for extracting segmentation map's. Based on the region-level difference annotations, we determine which regions undergo visual transitions, *i.e.*, RoEs, and construct corresponding global maps by fusing local maps from the current and next session images.

## 3.2 MODEL ARCHITECTURE

As illustrated in Fig. 3, our model is built upon a Diffusion Transformer (DiT) architecture, initialized from a video foundation model. We represent the interleaved input sequence as $S = (I_0, T_0, \ldots, T_{M-1}, I_M)$, where $T_i$ denotes the textual editing instruction at turn-$i$, and $I_i$ represents either an image or a segmentation mask.

As our focus is on the in-context image editing task, we optimize the model by maximizing the likelihood of the next image prediction:

$$\log p(S) = \sum_{i=1}^{M} \log p(I_i \mid I_0, \ldots, T_{i-1}, I_{i-1}) \tag{1}$$

where the conditional probability is modeled using flow-matching in the latent space, an objective commonly used in diffusion model for text-to-image (Rombach et al., 2022; Esser et al., 2024; Labs, 2024; Podell et al., 2023) and text-to-video (Singer et al., 2022; Wan et al., 2025; Hong et al., 2022; Seawead et al., 2025; Qu et al., 2025c) generation tasks. Each text instruction ($T_i$) and image ($I_i$) is encoded into latent tokens using a text encoder (*e.g.*, T5) and an image encoder (*e.g.*, VAE), respectively. The details about the text encoder and VAE are provided in the supplementary material.

**Learnable `<TURN>` Tokens**. We separate the interleaved input sequence $S$ by modality into two groups: $S = (I_0, T_0, \ldots, T_{M-1}, I_M) \rightarrow T = (T_0, T_1, ..., T_{M-1}); I = (I_0, ..., I_M)$. Their latent

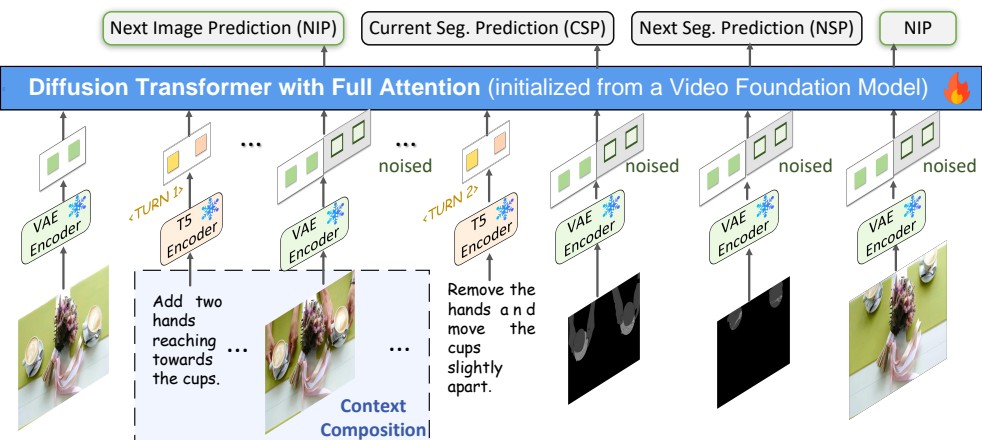

Figure 3: Model architecture. We apply a diffusion transformer framework (initialized from a video generative foundation model) with full attention to learn from the multimodal interleaved context, through three tasks (CSP, NSP, and NIP). Losses are only computed on noised tokens.

tokens are concatenated together. Since the number of text tokens at each turn may vary, we introduce $M$ special learnable tokens $\texttt{<TURN>}_i, i = 1, ..., M$ to mark the turn boundary, where $\texttt{<TURN>}_i$ is inserted before the latent tokens of $T_i$.

**Separate Text and Image Position Embedding**. We apply 1D RoPE (Su et al., 2024) to text tokens and 3D RoPE to image tokens. The starting positions are 0 for all dimensions. This separate RoPE design aligns with our pretrained MM-DiT model, where text and image tokens are positioned continuously. Position collisions are avoided as MM-DiT employs distinct weights for each modality, and the bias terms in the linear layers effectively act as modality-specific embeddings.

**Attention**. We employ two attention mechanisms in DiT and obtain two variants: (1) full attention over all tokens, as shown in Fig. 3, and (2) block-wise causal attention, where causality is enforced across blocks (*e.g.*, text or image) and bidirectional attention is applied within each block. Full attention enables comprehensive token interactions at a higher computational cost, while block-wise causal attention improves efficiency while maintaining causal structure. Additional details and discussions are provided in Appendix C.4.

**Condition on Clean Context**. We model the distribution of each image (except the first) using a diffusion loss, conditioned on an interleaved context. To enhance training efficiency, we concatenate the clean and noisy tokens of each image as model inputs, and apply an attention mask to ensure that each noisy image attends only to the clean representations of preceding images, as illustrated in Fig. 11.

### 3.3 CONTEXT COMPOSITION LEARNING

To facilitate effective ability transfer from segmentation modeling to image editing and generation, we unify image and segmentation modeling within a generative framework using the MSE-based diffusion loss in flow matching. Through interleaved context composition, our framework further unlocks multiple capabilities and supports a variety of corresponding tasks (see Fig. 12 for more details). Specifically, we augment Eqn. 1 by adding a random dropout operation $Rd$ on the context, as shown in equation:

$$\log p(S) = \sum_{i=1}^{M} \log p(F_i \mid Rd(I_0, T_1), Rd(T_{m0}, M_{00}), Rd(T_{m1}, M_{01}), \dots) \tag{2}$$

where $F_i$ can be either the target image, RoE mask[1] of the source image, or RoE mask of the target image. We ensure that the image or mask required to generate the target is always retained, while only the contextual images and texts are randomly dropped. The model is jointly learning three tasks:

---

[1]In implementation, we treat segmentation masks as RGB images, by replicating the mask across all three channels, and then encode them using the VAE encoder to obtain the latents.

- **Next Image Prediction (NIP)**. NIP is our primary in-context image editing task.

- **Current Segmentation Prediction (CSP)**. CSP enhances the model's *grounding* ability, enabling it to identify regions requiring edits while preserving consistency in other areas. This is particularly useful for local editing tasks such as removal, attribute changes, and replacements.

- **Next Segmentation Prediction (NSP)**. NSP improves the model's *controllable generation* by incorporating the next-frame segmentation map into the context, aiding in dynamic layout adjustments for scenarios like shape changes and movements.

By randomly combining different contexts and tasks, the model learns essential abilities such as grounding, controllable generation, and multi-concept composition, enabling versatile in-context image editing.

## 4 EXPERIMENTS

### 4.1 IMPLEMENTATION DETAILS

**Data**. Through the proposed scalable data construction pipeline, we collect and annotate about 10M session instances, with the number of images in each session from 2 to 20. For each session data, we consider RoE map with a probability of 80%. We apply a context drop rate with 20%, 70%, and 70%, to the current frame, current RoE map, and next RoE map, respectively. During inference, the sampling step is set to 50, the classifier-free guidance scale is set to 10. Using the proposed data construction pipeline, we collect and annotate about 10M session instances, each containing 2 to 20 images. During training, a RoE map is included with an 80% probability for each session. We apply context dropout rates of 20%, 70%, and 70% to the current frame, current RoE map, and next RoE map, respectively, with dropout applied independently at each turn. We use 50 sampling steps and set the classifier-free guidance scale to 10.

**Model.** We initialize our model with the weights of our in-house MM-DiT (3B and 7B), pre-trained on text-to-video tasks and architecturally similar to (Seawead et al., 2025; Kong et al., 2024). The 3B and 7B variants are optimized on session data for 15k and 40k steps, consuming approximately 30 and 150 hours on 256 H100 GPUs, respectively.

### 4.2 MULTI-TURN SESSION IMAGE EDITING BENCHMARK

Existing benchmarks (Zhang et al., 2023a; Basu et al., 2023; Sheynin et al., 2024), such as MagicBrush (Zhang et al., 2023a), are constrained to basic editing operations, such as addition, replacement, removal, attribute modification, and background changes, and thus fall short of meeting practical user needs. Moreover, MagicBrush supports only up to three editing turns per session, with each turn treated in isolation, further diverging from real-world editing workflows. To address these limitations, we propose **MSE-Bench** (Multi-turn Session image Editing Benchmark), which comprises 100

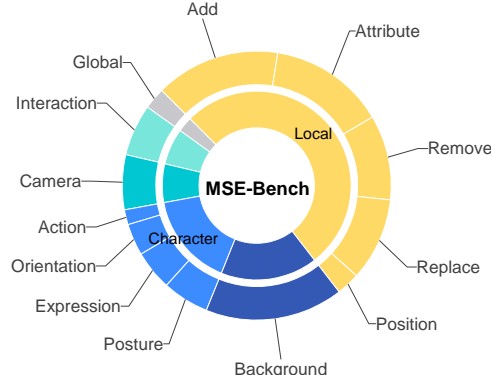

Figure 4: Category distribution of MSE-Bench. "others" includes expression, orientation, position, global, and action change.

test instances, each featuring a coherent five-turn editing session. MSE-Bench *expands the range of editing categories* to include more complex and realistic scenarios such as posture adjustment, object interaction, and camera view changes, as shown in Fig. 4. To better reflect user intent and practical applications, we also incorporate *aesthetic* considerations into the construction, encouraging progressive visual enhancement across turns.

For each editing instruction, multiple generated images may satisfy the user's request. Consequently, our benchmark does not provide ground-truth images. Instead, we use GPT-4o to evaluate whether the generated image successfully follows the instructions and remains consistent with the input image. The final score for each turn is computed by averaging the success rates across all samples.

Table 1: Performance comparison on MagicBrush (Zhang et al., 2023a) (multi-turn) for consistency (DINO and CLIP-I) and prompt following (CLIP-T). SFT means we carry out supervised fine-tuning. * denotes the use of context across all preceding turns. Entries by gray denote proprietary models.

| Method | Turn-1 | | | Turn-2 | | | Trun-3 | | |
|---|---|---|---|---|---|---|---|---|---|
| | DINO | CLIP-I | CLIP-T | DINO | CLIP-I | CLIP-T | DINO | CLIP-I | CLIP-T |
| Instruct-Pix2Pix (Brooks et al., 2023) | 0.514 | 0.727 | 0.270 | 0.397 | 0.674 | 0.268 | 0.335 | 0.646 | 0.263 |
| MagicBrush (Zhang et al., 2023a) | 0.826 | 0.901 | 0.278 | 0.756 | 0.863 | 0.277 | 0.718 | 0.834 | 0.271 |
| HQEdit (Hui et al., 2024) | 0.522 | 0.696 | 0.259 | 0.441 | 0.659 | 0.248 | 0.397 | 0.637 | 0.238 |
| UltraEdit (Zhao et al., 2024) | 0.755 | 0.852 | 0.289 | 0.706 | 0.827 | 0.278 | 0.683 | 0.810 | 0.266 |
| ICEdit (Zhang et al., 2025b) | 0.853 | 0.922 | 0.281 | 0.780 | 0.882 | 0.278 | 0.731 | 0.852 | 0.272 |
| OmniGen (Zhang et al., 2025b) | 0.874 | 0.924 | 0.273 | 0.718 | 0.851 | 0.264 | 0.586 | 0.786 | 0.261 |
| OmniGen2 (Wu et al., 2025b) | 0.863 | 0.919 | 0.285 | 0.777 | 0.869 | 0.280 | 0.716 | 0.832 | 0.278 |
| Step1X-Edit (Liu et al., 2025) | 0.852 | 0.915 | 0.288 | 0.785 | 0.875 | 0.286 | 0.743 | 0.840 | 0.277 |
| Bagel (Deng et al., 2025) | 0.845 | 0.912 | 0.286 | 0.767 | 0.873 | 0.292 | 0.723 | 0.844 | 0.286 |
| Bagel* (Deng et al., 2025) | 0.847 | 0.914 | 0.287 | 0.729 | 0.858 | 0.295 | 0.684 | 0.823 | 0.287 |
| FLUX.1-Kontext (dev) (Batifol et al., 2025) | 0.858 | 0.917 | 0.288 | 0.757 | 0.863 | 0.296 | 0.691 | 0.818 | 0.291 |
| Qwen-Image-Edit (Wu et al., 2025a) | 0.827 | 0.900 | 0.292 | 0.745 | 0.856 | 0.292 | 0.697 | 0.819 | 0.287 |
| GPT Image 1* (OpenAI, 2025a) | 0.805 | 0.875 | 0.293 | 0.708 | 0.820 | 0.300 | 0.666 | 0.789 | 0.292 |
| Nano Banana* (DeepMind & Gemini, 2025) | 0.886 | 0.933 | 0.287 | 0.811 | 0.896 | 0.294 | 0.773 | 0.867 | 0.291 |
| Ours* (3B) | 0.822 | 0.895 | 0.273 | 0.733 | 0.850 | 0.272 | 0.676 | 0.827 | 0.267 |
| Ours* (3B) + SFT | 0.852 | 0.917 | 0.283 | 0.739 | 0.861 | 0.291 | 0.667 | 0.814 | 0.290 |
| Ours* (7B) | 0.838 | 0.906 | 0.272 | 0.721 | 0.848 | 0.272 | 0.645 | 0.804 | 0.271 |
| Ours* (7B) + SFT | **0.891** | **0.937** | 0.283 | **0.817** | **0.895** | 0.289 | **0.775** | **0.861** | 0.286 |

## 4.3 COMPARISON WITH STATE-OF-THE-ARTS

We evaluate our model on two multi-turn image editing benchmarks: MagicBrush (Zhang et al., 2023a) and our proposed MSE-Bench.

**MagicBrush**. Given its support for multi-turn editing, high-quality manual annotations, and close alignment with real-world editing needs, we first adopt MagicBrush to evaluate our method and compare against baselines.

Tab. 1 reports quantitative results across three standard evaluation metrics: DINO, CLIP-I, and CLIP-T. First, our model, trained solely on interleaved video data, achieves performance comparable to SOTA methods UltraEdit and OmniGen, which rely on pairwise editing data, highlighting video data as a natural and effective source for image editing tasks. Second, with supervised fine-tuning on editing-oriented data, our method outperforms nearly all metrics, demonstrating that interleaved video data complements existing data creation approaches. Lastly, our model's advantages become increasingly evident with more edit turns, showcasing the benefits of learning from contextual video data.

**MSE-Bench**. Tab. 2 presents the multi-turn editing success rates as evaluated by GPT-4o. In this setup, the generated image at turn-$i$ serves as the input for editing at turn-$i + 1$. Consequently, failure at any turn propagates to subsequent turns. Existing academic methods perform poorly, with a success rate of **< 2%** at turn-5. In contrast, our method achieves a **25%** success rate at turn-5, demonstrating the advantages of our model and the use of native video data. However, our approach still falls short compared to proprietary models like GPT-4o, which benefit from significantly larger training datasets and model sizes. Even so, GPT-4o achieves only a **62.7%** success rate, highlighting the long-term value of our proposed benchmark for advancing multi-turn editing.

## 4.4 IN-DEPTH ANALYSIS

**In-Context Editing Mitigates Artifact Accumulation**. Artifact accumulation, where artifacts become more pronounced with increasing editing turns, is a common issue in multi-turn editing (Sheynin et al., 2024). We observe this phenomenon as well (upper part of Fig. 6) when using our model as a single-turn editing method, *i.e.*, without incorporating context from previous turns. However, when all contexts are included as input, no artifacts are observed (lower part of Fig. 6).

**Impact of Segmentation Prediction and Generation.** As shown in Tab. 3, training with segmentation and generation as context enhances both consistency and multi-turn editing success rate. Notably, the substantial gain in consistency on MagicBrush (Zhang et al., 2023a) demonstrates the effectiveness of segmentation modeling, especially under the CoE strategy (CS → NS → I).

Table 2: Performance comparison on MSE-Bench (editing success rate evaluated by GPT-4o). * denotes the use of context across all preceding turns. Entries by gray denote proprietary models.

| Method | GPT-4o Evaluation | | | | |
| --- | --- | --- | --- | --- | --- |
| | Turn-1 | Turn-2 | Turn-3 | Turn-4 | Turn-5 |
| Instruct-Pix2Pix (Brooks et al., 2023) | 0.520 | 0.130 | 0.110 | 0.083 | 0.060 |
| MagicBrush (Zhang et al., 2023a) | 0.707 | 0.300 | 0.213 | 0.170 | 0.087 |
| HQEdit (Hui et al., 2024) | 0.477 | 0.177 | 0.140 | 0.113 | 0.077 |
| UltraEdit (Zhao et al., 2024) | 0.673 | 0.230 | 0.173 | 0.113 | 0.067 |
| ICEdit (Zhang et al., 2025b) | 0.633 | 0.340 | 0.257 | 0.163 | 0.090 |
| OmniGen (Xiao et al., 2024) | 0.847 | 0.223 | 0.170 | 0.140 | 0.083 |
| OmniGen* (Xiao et al., 2024) | 0.853 | 0.188 | 0.160 | 0.125 | 0.065 |
| OmniGen2 (Wu et al., 2025b) | 0.847 | 0.393 | 0.327 | 0.263 | 0.133 |
| Step1X-Edit (Liu et al., 2025) | 0.937 | 0.540 | 0.420 | 0.300 | 0.140 |
| Bagel (Deng et al., 2025) | 0.967 | 0.650 | 0.613 | 0.550 | 0.413 |
| Bagel* (Deng et al., 2025) | 0.963 | 0.630 | 0.567 | 0.473 | 0.300 |
| FLUX.1-Kontext (dev) (Batifol et al., 2025) | 0.950 | 0.670 | 0.623 | 0.573 | 0.440 |
| Qwen-Image-Edit (Wu et al., 2025a) | **0.980** | **0.737** | **0.667** | 0.613 | 0.430 |
| GPT Image 1 (OpenAI, 2025a) | 0.963 | 0.690 | 0.673 | 0.637 | 0.557 |
| GPT Image 1* (OpenAI, 2025a) | 0.967 | 0.707 | 0.700 | 0.697 | 0.640 |
| Nano Banana (DeepMind & Gemini, 2025) | 0.987 | 0.773 | 0.753 | 0.727 | 0.627 |
| Nano Banana* (DeepMind & Gemini, 2025) | 0.997 | 0.773 | 0.757 | 0.730 | 0.643 |
| Ours* (3B) | 0.913 | 0.450 | 0.393 | 0.300 | 0.210 |
| Ours* (3B) + SFT | 0.913 | 0.533 | 0.497 | 0.443 | 0.330 |
| Ours* (7B) | 0.837 | 0.517 | 0.463 | 0.400 | 0.350 |
| Ours* (7B) + SFT | 0.950 | 0.693 | **0.667** | **0.617** | **0.487** |

Table 3: **Impact of segmentation (seg.) prediction and generation as context during training and inference on consistency** (CLIP-I and DINO on MagicBrush) **and success rate** (evaluated by GPT-4o). I: image generation. CS: current segmentation generation. NS: next segmentation generation. (This ablation study was conducted using an intermediate checkpoint, so the reported numbers may not be directly comparable to those in other tables. )

| Train | Inference | MagicBrush (CLIP-I) | | | MagicBrush (DINO) | | | MSE-Bench (Success Rate by GPT-4o) | | | | |
| --- | --- | --- | --- | --- | --- | --- | --- | --- | --- | --- | --- | --- |
| | | Turn-1 | Turn-2 | Turn-3 | Turn-1 | Turn-2 | Turn-3 | Turn-1 | Turn-2 | Turn-3 | Turn-4 | Turn-5 |
| w/o Seg. | I | 0.875 | 0.824 | 0.784 | 0.765 | 0.663 | 0.592 | 0.847 | 0.473 | 0.337 | 0.177 | 0.113 |
| w/ Seg. | I | 0.880 | 0.832 | 0.797 | 0.786 | 0.680 | 0.604 | **0.887** | 0.520 | 0.327 | 0.183 | 0.103 |
| w/ Seg. | CS → I | 0.886 | 0.832 | 0.801 | 0.797 | 0.687 | 0.622 | 0.873 | **0.590** | **0.407** | **0.260** | **0.173** |
| w/ Seg. | NS → I | 0.889 | 0.840 | 0.815 | 0.807 | 0.711 | 0.661 | 0.837 | 0.487 | 0.323 | 0.197 | 0.117 |
| w/ Seg. | CS → NS → I | **0.890** | **0.847** | **0.823** | **0.814** | **0.724** | **0.679** | 0.867 | 0.523 | 0.367 | 0.190 | 0.110 |

**Impact of Context.** Table 4 highlights the impact of context in multi-turn image editing. In Turn-1, where no prior context exists, adding a dummy context—comprising the original image and an instruction, In Turn-2 and Turn-3, where editing instructions and ground-truth images from previous turns are provided as context, adding a dummy context results in minimal improvements. "generate the same image," prepended before Turn-1—significantly improves performance. The L1 and L2 distances are nearly halved, indicating greater consistency between the generated image and the original image in unchanged areas, as these distances are measured pixel-wise. This is expected, as the existing context already provides sufficient information. These findings underscore the critical role of context in multi-turn image editing tasks.

Table 4: **Impact of context on multi-turn image editing with MagicBrush.** The "Dummy-Context" includes the original image and the instruction, "generate the same image." "History" refers to providing previous turns' ground-truth images as context. Results show that performance significantly improves when a reasonable context is included, emphasizing the importance of context in multi-turn image editing.

| Method | L1↓ | L2↓ | DINO↑ | CLIP-I↑ | CLIP-T↑ |
| --- | --- | --- | --- | --- | --- |
| | | | Turn-1 | | |
| w/o Context | 0.155 | 0.063 | 0.814 | 0.894 | 0.277 |
| Dummy-Context | 0.086 | 0.031 | 0.850 | 0.913 | 0.277 |
| | | | Turn-2 | | |
| w/o Context | 0.159 | 0.067 | 0.834 | 0.902 | 0.279 |
| History | 0.099 | 0.038 | 0.845 | 0.909 | 0.278 |
| Dummy-Context | 0.087 | 0.033 | 0.869 | 0.922 | 0.280 |
| | | | Turn-3 | | |
| w/o Context | 0.164 | 0.071 | 0.851 | 0.904 | 0.273 |
| History | 0.088 | 0.034 | 0.878 | 0.923 | 0.273 |
| Dummy-Context | 0.088 | 0.034 | 0.895 | 0.929 | 0.272 |

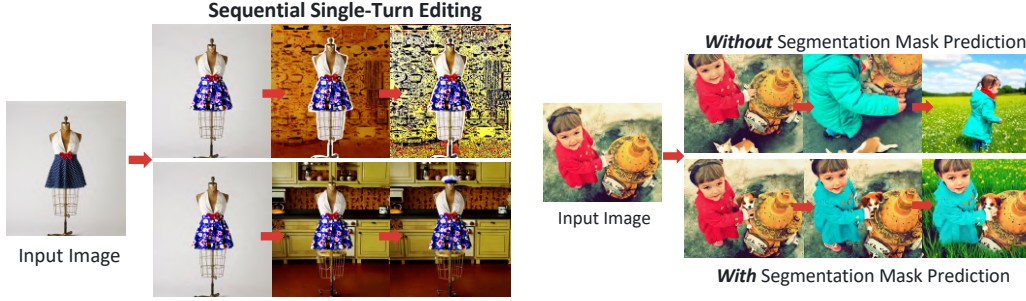

**Sequential Single-Turn Editing**

Input Image

**In-Context Editing**

Figure 6: In-context editing mitigates artifact accumulation issue in sequential single-turn editing.

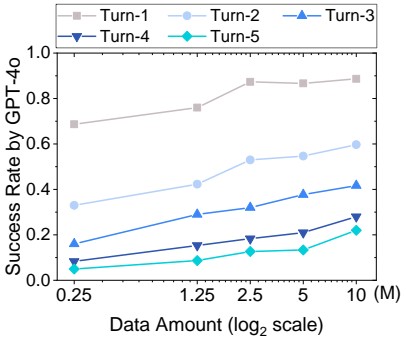

*Without* Segmentation Mask Prediction

Input Image

*With* Segmentation Mask Prediction

Figure 7: Subject position shift can be addressed by predicting segmentation mask first.

**Scalability.** Fig. 5 illustrates the editing success rate as a function of training data size. While the success rate at Turn-1 begins to saturate at 2.5M training samples, the success rate at later turns (*e.g.*, Turn-4 and Turn-5) exhibits a nearly log-linear increase with more training data. These results demonstrate the scalability of both our model and data construction pipeline.

**Training on Native Video Data Introduces Addressable Subject Position-Shift.** A key challenge when training on videos is the potential for subject position shifts across editing turns, as illustrated in the upper part of Fig.7. This issue arises from the natural movement of subjects over time in videos. However, incorporating segmentation prediction—where the model first predicts a mask before generating the target image—mitigates this drifting effect (see lower part of Fig.7). The segmentation mask enforces consistency in unedited regions, thereby reducing positional drift.

Figure 5: Editing success rates in 5 turns at various data scales.

**Effectiveness of Our Video Sequence Data**. Table 5 demonstrates the impact of incorporating our video sequence data. Using the same pretrained model, training with our video sequence data increases success rates by **16.4%** and **21.0%** on Turn-1 and Turn-5, respectively, compared to training solely on specialized pairwise image editing data (Wei et al., 2024). The highest performance is achieved by first pretraining on our video sequence data, followed by supervised fine-tuning (SFT) on pairwise data, underscoring the effectiveness of our data for continual pretraining.

Table 5: Ablation study on MSE-Bench (GPT-4o evaluated success rate), to assess the impact of our video sequence data.

| Training Data | Turn-1 | Turn-2 | Turn-3 | Turn-4 | Turn-5 |
|---|---|---|---|---|---|
| pairwise | 0.723 | 0.263 | 0.123 | 0.033 | 0.010 |
| sequence | **0.887** | 0.597 | 0.417 | 0.280 | 0.220 |
| sequence → pairwise | 0.880 | **0.647** | **0.483** | **0.370** | **0.250** |

### 4.5 APPLICATIONS

Fig. 1 showcases several emerging capabilities that arise when training our model exclusively on video data. Notably, these abilities seem to develop implicitly, as they differ from the model's explicit training objectives:

- **Controllable Editing:** By including the segmentation mask of the region of interest in the context, users can achieve controllable editing by modifying the segmentation mask.
- **Multi-Concept Composition:** The model demonstrates the ability to compose multiple concepts together, even without explicit composition training data—a surprising emergent capability.
- **Story Generation:** Leveraging the consistent and extended context in video data, the model can generate coherent frames for storytelling through in-context editing.
- **Chain-of-Editing:** Each multi-turn editing session functions as a multimodal chain of thought, where the model interprets editing instructions, identifies regions of interest, generates RoI masks,

produces target images, and iterates the process. Our model reveals the potential of video data in modeling multimodal chains of thought.

## 5 CONCLUSION

In this work, we explore the research question: "Can an in-context image editing model be learned solely from videos?" To address this, we propose a learning framework that enables context-aware image generation directly from native videos. We introduce a scalable data construction pipeline that transforms videos into contextual multimodal sequences, comprising sparsely sampled frames, textual visual transition descriptions, and segmentation masks of regions of interest. To model this multimodal sequence, we train a DiT model using three proxy tasks: next-image prediction, current segmentation prediction, and next-segmentation prediction. Experimental results demonstrate that our model, trained exclusively on videos, exhibits strong in-context image editing capabilities and achieves state-of-the-art performance on multiple multi-turn image editing benchmarks. Additionally, our model showcases emerging abilities such as controllable editing, multi-concept composition, story generation, and multimodal chain-of-thought, highlighting the untapped potential of video data and the effectiveness of our proposed framework.

### ETHICS STATEMENT

Our work on scalable, context-aware image editing has the potential to democratize creative tools, enhance accessibility, streamline media production, and advance intuitive human-AI collaboration. However, it also raises important concerns, including the risk of misuse for misinformation or manipulation, privacy issues from large-scale video data, potential biases in generated content, job displacement in creative industries, and increased environmental impact due to computational demands. Addressing these challenges will require careful dataset curation, privacy safeguards, bias mitigation, responsible deployment practices, and ongoing engagement with diverse stakeholders.

### REPRODUCIBILITY STATEMENT

We have taken several steps to ensure the reproducibility of our work. A link ([https://vincie2025.github.io/](https://vincie2025.github.io/)) to the source code is provided, enabling replication of our implementation. The main text and appendix together provide comprehensive descriptions of the model design, training procedure, and evaluation protocol. Details on the dataset construction and preprocessing pipeline are presented in the appendix. These resources collectively ensure that readers can reproduce and validate our experimental results.

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

CONTENTS

## A    THE USE OF LARGE LANGUAGE MODELS

We acknowledge that large language models (LLMs) were employed to assist in the preparation of this manuscript. Their use was restricted to grammar checking, language refinement, and enhancing clarity and fluency of the text. In addition, LLMs were applied in a limited capacity to support minor debugging and syntactic corrections of code snippets.

## B    ADDITIONAL RELATED WORK

**Image Editing**. Building on advances in foundational image generation models (Huang et al., 2025; Ramesh et al., 2022; Saharia et al., 2022; Esser et al., 2024), image editing has achieved remarkable progress. Techniques now enable a wide range of edits, including zero-shot editing (Li et al., 2024; Huang et al., 2023; Wu & De la Torre, 2023; Han et al., 2024a; Zhou et al., 2025b; Chen & Huang, 2023; Zhou et al., 2025a), changing object classes (Kim et al., 2022; Xu et al., 2023; Ackermann & Li, 2022; Yang et al., 2023c; Tsaban & Passos, 2023; Gholami & Xiao, 2023; Brack et al., 2024; Nie et al., 2023) and faces (Ding et al., 2023), free-form text-based modifications (Brooks et al., 2023; Hertz et al., 2022; Lin et al., 2023; Dong et al., 2023; Zhang et al., 2023b; Kawar et al., 2023; Guo & Lin, 2024; Zhang et al., 2024; Sheynin et al., 2024; Wei et al., 2024; Shi et al., 2024a; Wang et al., 2023a; Li et al., 2023; Mirzaei et al., 2024; Miyake et al., 2025), mask-based edits (Wang et al., 2023b; Xie et al., 2023; Couairon et al., 2022; Zou et al., 2024; Mao et al., 2024), point dragging (Mou et al., 2023; Shin et al., 2024; Liu et al., 2024a; Lu et al., 2024; Choi et al., 2025), and reference image-guided transformations (Song et al., 2023; Goel et al., 2023; Yang et al., 2023a). A series of recent works (Yang et al., 2023b; Wu et al., 2023; Xiao et al., 2024; Najdenkoska et al., 2024; Sun et al., 2024) enables edits conditioned on multiple text and images. Our work focuses on in-context image editing (OpenAI, 2025b), where edits are conditioned on a contextual sequence of text and *previously generated* images. Moreover, we explore learning from native video data, unlike existing methods that use hand-crafted synthesized data.

## C    IMPLEMENTATION DETAILS

### C.1    DATA DETAILS

The training videos are sourced from a wide spectrum of domains, including stock footage, films, documentaries, etc. We split the raw videos into both single-shot clips and multi-shot scene videos. We also pre-process the raw videos by using different filtering strategies to keep high-quality videos, including logo detection, black border detection, and aesthetic estimation.

As described in Sec.3.1, we adopt two frame sampling strategies: equal-interval sampling and fixed-frame sampling. As illustrated in Fig.8, these approaches jointly ensure both the diversity and temporal stability of visual dynamics—two key factors for effective training of in-context image editing models.

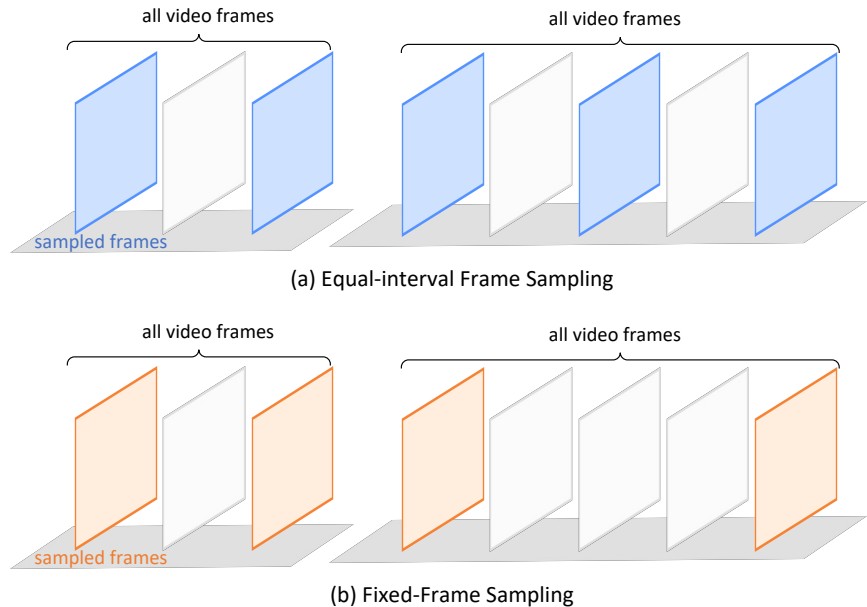

Figure 8: Two ways of frame sampling: (a) equal-interval sampling and (b) fixed-frame sampling.

## C.2 Visual Transition Annotation

---

**Instruction for Visual Transition Annotation**

Imagine that you are an image editing assistant who wants to edit the first image to the second image. I will provide you two frames from a video clip as the source and target images. The caption of the raw video clip is: { }

Your task is to summarize how you intend to achieve this image editing task by providing detailed but brief text instructions, from the following guidelines:
1. Understand the two images first, and describe the two frames in detail and coherently. Please include the details of the environment, main subjects, their appearances, and main features.
2. Describe the main characters and objects and their appearances. Do not mention the real name entities. Follow the format such as: {"char1": "a woman with blonde hair wearing a red jacket", "char2": "a girl wearing a floral dress", "obj1": "a green apple", ...}
3. Highlight the semantic and visual differences between the two images in detail.
4. Provide only factual descriptive differences based on observable content. Avoid words or phrases that suggest speculation or assumptions, such as "likely", "possibly", or "appear to".
5. Avoid elliptical referential pronouns, such as "the same, frame 1, frame 2, the first image, the second image, ... ".

An editing instruction should include:
1. main character change, including appearance, disappearance, position, action, expression, pose, orientation, ... (*e.g.*, "make the person smile")
2. object change, including appearance, disappearance, position, count, relationship, layout, ... (*e.g.*, "add a dog beside the person")
3. attribute change, including color, texture, material, shape, size, depth, dynamics, ... (*e.g.*, "make the person's hair red")
4. interaction change, including the interaction between characters, objects, and the environment. (*e.g.*, "make the person hold the dog")
5. global change, including background, atmosphere, environment, style, weather, season, lighting, ... (*e.g.*, "make the weather dark")
6. camera change, including orbiting, dolly-in, dolly-out, pan-left, pan-right, tilt-up, tilt-down.
7. others

Output Format: You should output a json file to include the following information:
Frame1 Caption: <describe the first image/frame, characters and objects in detail>
Frame2 Caption: <describe the second image/frame, characters and objects in detail>
Character Change: <the detailed character and attribute change>
Object Change: <the detailed object and attribute change>
Global Change: <the detailed global change>
Camera Change: <the detailed camera change>
Other Change: <the detailed other change>
Summary Change: <a comprehensive but brief user editing instruction to achieve the editing>
Your output should be a JSON file in one row (without any format), which looks like:
{"frame1_caption": {"scene": str, "char1": str, "char2": str, ..., "obj1": str, "obj2": str, ...}, "frame2_caption": {"scene": str, "char1": str, "char2": str, ..., "obj1": str, "obj2": str, ...}, "character_change": {"char1": str, "char2": str, ...}, "object_change": {"obj1": str, "obj2": str, ...}, "global_change": str, "camera_change": str, "other_change": str, "summary_change": str}

---

To bridge the semantic gap between two sampled frames, we use our in-house LMM to annotate visual transitions, as introduced in Sec.3.1. The instruction used during annotation is shown above, and Fig. 10 presents example annotations to illustrate their quality.

## C.3 Segmentation Mask Annotation and RoE Construction

The proposed visual transition annotation framework leverages an LMM to generate multi-level annotations, ranging from local concepts to global scene descriptions. As illustrated in Fig.2, we first use character and object descriptions from the source and target frames as query inputs to GroundingDINO(Liu et al., 2024b) to obtain object detection results. These detections are then passed to SAM 2 (Ravi et al., 2024) to extract segmentation masks for the corresponding local concepts. Guided by the annotated local changes, we identify and fuse the objects or characters undergoing transitions to construct the final RoEs.

Turn 1: Replace the crescent moon and stars with a smiling sun to change the time of day to daytime.

Turn 1: Remove the rocky cliff and mist, change the background to an ocean - side beach with a hazy sky.
Turn 2: Swap the positions of the two women, let them stand sideways, and make them face the camera more directly.

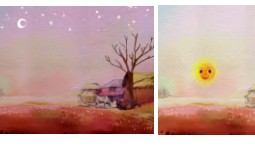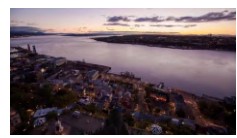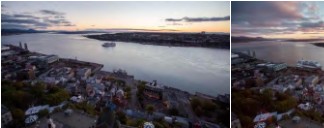

Turn 1: Remove the rocky cliff and mist, change the background to an ocean - side beach with a hazy sky. Swap the positions of the two women, let them stand sideways, and make them face the camera more directly

Turn 1: Change the man's facial expression from neutral to an open - mouthed expression as if speaking or exclaiming
Turn 2: Remove the man from the image and close the door.

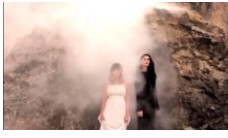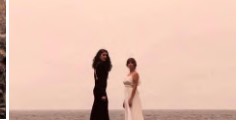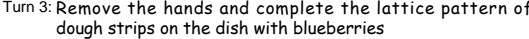

Turn 1: Change the background landscape to show more greenery, smaller water bodies, and add some buildings near the shoreline in the distance. Turn the man's head slightly to the right.

Turn 1: Remove the hand and add blueberries evenly spread over the dough.
Turn 2: Add a pair of hands creating a lattice - pattern with dough strips on top of the blueberries in the baking dish
Turn 3: Remove the hands and complete the lattice pattern of dough strips on the dish with blueberries

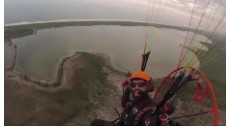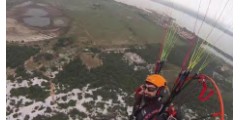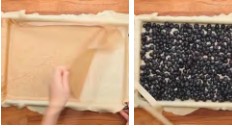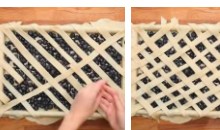

Turn 1: Change the visible part of the man's face to show more of the eyes and forehead, add hair on the forehead, and add a red - outlined white mark on the forehead
Turn 2: Pan down the camera to focus on the man's nose and mouth area and move the red - outlined white patch from the forehead to the lower lip
Turn 3: Zoom out to show the full face of the man, add hair, change the framing to include a plain background, and change the expression to neutral

Turn 1: Transform the simple fox - like sketch into a detailed female character with fox - like features performing a dance move
Turn 2: Change the female character's dance pose from having one arm raised and one leg lifted to having both arms extended and one leg forward
Turn 3: Change the female character's pose to standing upright with arms raised and add wings behind her

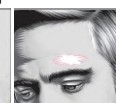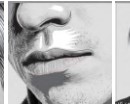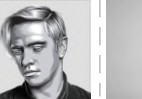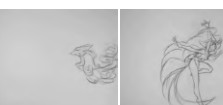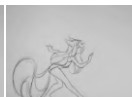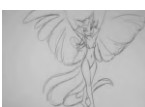

Figure 9: Examples (1/2) of visual transition annotation performed by our in-house large multimodal model.

Turn 1: Stop the man's hand - gesturing and close his mouth 1,
Turn 2: Add a curved stick to the man's left hand and make him gesture with his right hand.
Turn 3: Change the man's hand gesture from a general gesture to a rock - on hand gesture with the arm raised higher

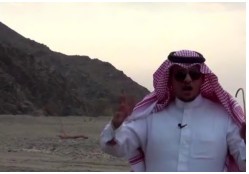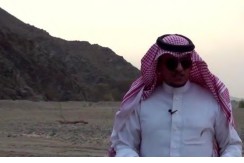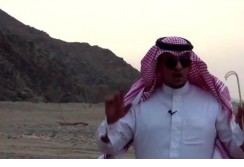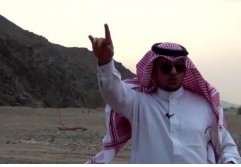

Turn 1: Change the boy's action to running with one arm extended towards the basketball and move the basketball to in front of the boy on the ground, and change the boy's orientation to face more towards the left.
Turn 2: Change the boy's action to standing upright and looking forward, move the basketball to the boy's right hand, and add a sun on the right side of the sky.
Turn 3: Add a black X on the boy's shorts and change his pose to holding the basketball up to his face

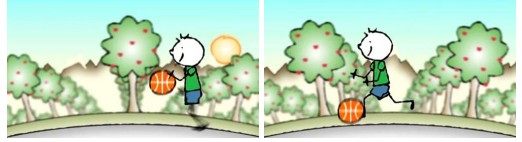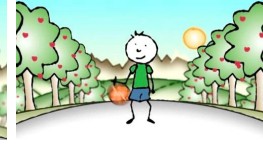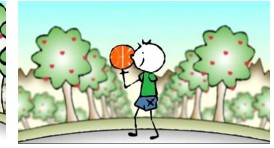

Turn 1: Remove the two women and add a white armchair with a blanket, a small black round table, a floor lamp with a white shade, and a potted plant.
Turn 2: Add a woman with red hair, wearing a yellow short - sleeved shirt and black pants, standing and facing away from the camera with her right hand raised slightly to the room scene.
change the woman's action from walking and gesturing to standing and touching the patterned curtain with both hands
Turn 3: Add a woman with blonde hair sitting on the armchair, holding a white cup and raising her hand.
Turn 4: Change the first woman's action to standing and holding a white cup and smiling. Add two white cups, one in each woman's hand.

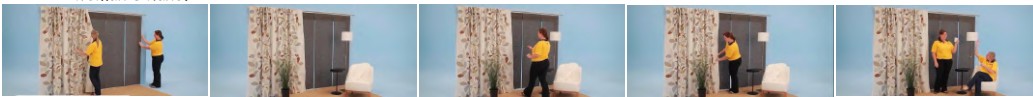

Turn 1: Move the man closer to the SUV such that he is opening the rear door with his right hand, and change the SUV's rear door to be open.
Turn 2: Edit the image to transition the man's position from standing outside the rear door of the SUV to being partially inside the vehicle, bent over.
Turn 3: Remove the man getting into the SUV and close the rear door.
Turn 4: Move the white SUV further down the path and close the rear right door.

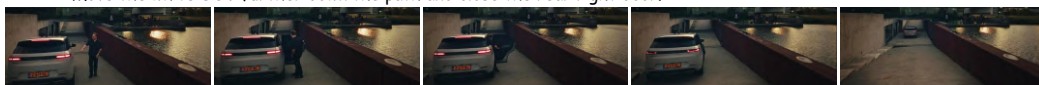

Figure 10: Examples (2/2) of visual transition annotation performed by our in-house large multimodal model.

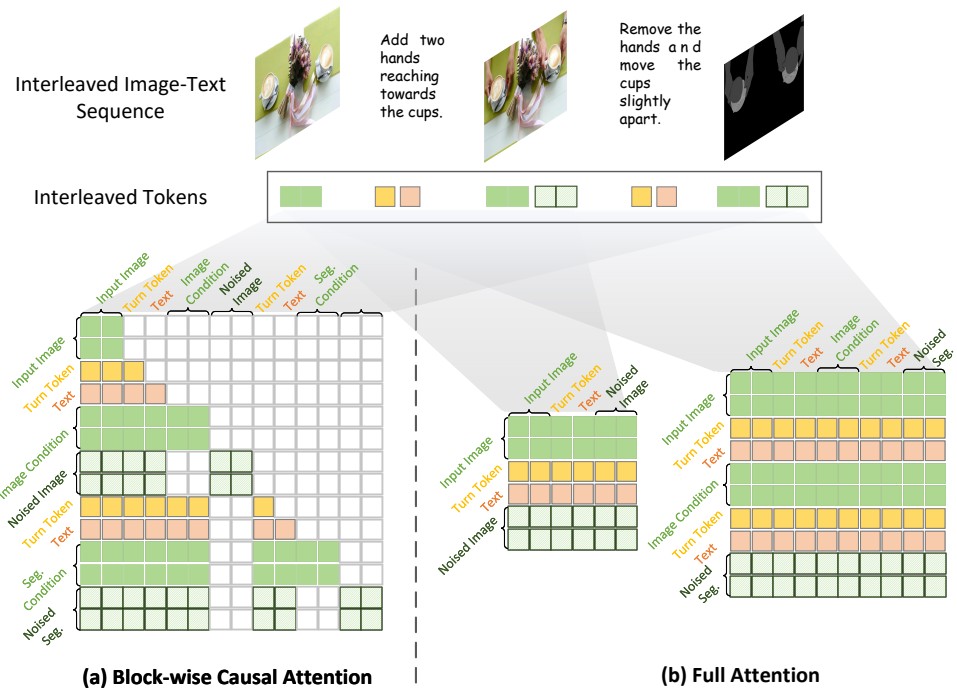

Figure 11: Implementation of (a) block-wise causal attention and (b) full attention.

## C.4 MODEL ARCHITECTURE

**Variational Autoencoder**. Following prior work (Yu et al., 2023), we adopt the encoder in a pretrained VAE to embed each image into the latent space separately for efficient computation. Specifically, it compress raw pixels with shape $(H, W, 3)$ into a $(h, w, c)$-shape latent representation, with downsampling ratios as $d_h = \frac{H}{h}$ and $d_w = \frac{W}{w}$ for height and width, respectively, and the latent channel $c$. The decoder in VAE aims to transform latent representations generated by the DiT back into the pixel space during inference.

**Text Encoder**. We employ the pretrained Flan-T5 as the text encoder to separately encode the prompt in each turn, and then concatenate all the embedding with inserting turn embeddings in between. Specifically, to make the model better discriminate different turns, we define a special turn token $<\texttt{TURN}>_i$ for the $i$-th turn, and introduce a learnable turn embedding for each one, which is inserted before the prompt embedding in the $i$-th turn.

**Full Attention and Block-wise Causal Attention**. We show the comparison between full attention and block-wise causal attention, and the condition strategy of clean context in block-wise causal attention, in Fig. 11.

## C.5 COMPOSITION OF INPUT CONDITIONS AND OUTPUT

In Fig. 12, we enumerate all seven context compositions supported by our method, detailing the interleaved input conditions, the corresponding outputs, the learning objectives, and the specific capabilities unlocked by each composition.

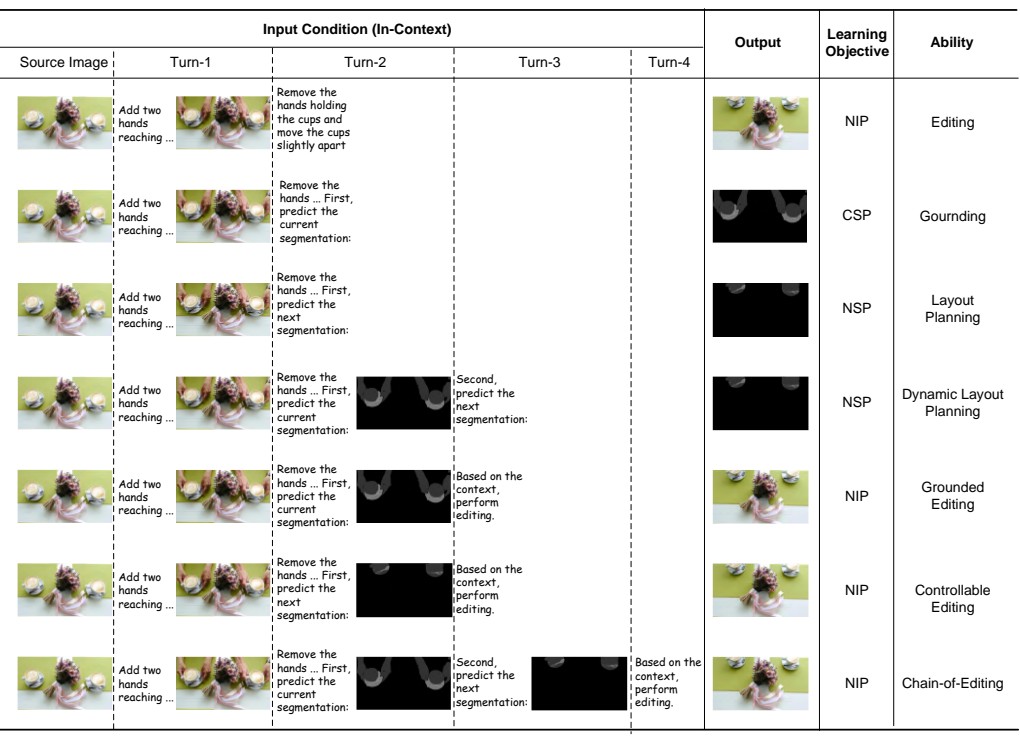

Figure 12: Context composition supported by our method.

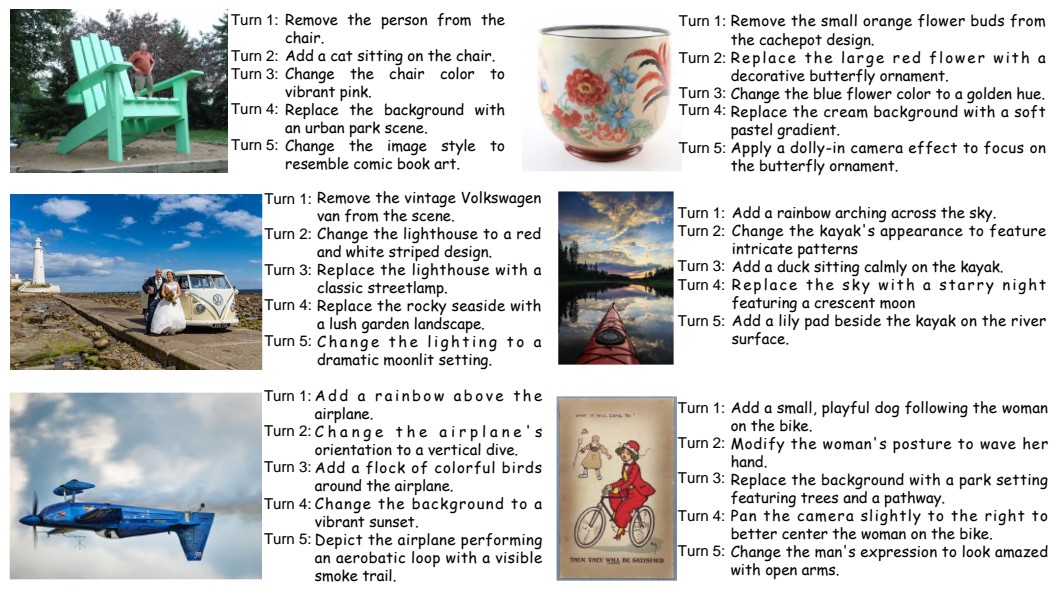

Figure 13: Multi-turn image editing examples of MSE-Bench.

## C.6 DETAILS OF MSE-BENCH

> **Instruction for Evaluation of Multi-turn (turn-i) Image Editing on MSE-Bench**
>
> Assume you are an expert in evaluating multi-turn image editing. In this task, a user interacts with an image editing system across multiple turns. At the first turn, the user provides a source image and an editing prompt. The system returns the edited image. In each subsequent turn, the user supplies a new prompt, and the system generates a new image based on the output from the previous turn. Your goal is to evaluate how successfully the editing instruction of the LAST turn has been executed.
> You will be given user editing prompts and images: the first image is the original source image, and the next are the edited results from each turn for each prompt. You should focus more on the last prompt and the last edited image, but you may also consider the previous prompts and images as context.
> The user editing prompts are: {}
>
> Please follow these evaluation rules:
> 1. Last-turn Evaluation: For the last turn, you should first assess the result based on two criteria by giving a reason: 1) prompt_following, does the last edited image fulfill the last user's editing prompt? 2) consistency: Are the untouched parts of the last result image consistent with the input reference (the source image at the first turn, or the result image at the previous turn)?
> 2. Scoring: Based on the reason, you assign scores for "prompt_following" and "consistency".
> From scale 0 to 10:
> A "prompt_following" score from 0 to 10 will be given based on the editing success of prompt following. (0 indicates that the scene in the last edited image does not follow the last editing instruction at all. 10 indicates that the scene in the last edited image follow the last editing instruction perfectly.)
> A "consistency" score from 0 to 10 will rate the degree of overediting in the last edited image. (0 indicates that the scene in the last edited image is completely different from the original. 10 indicates that the last edited image can be recognized as a minimally edited yet effective version of the original.)
> 3. Return your results in a JSON structure, following this format:
> {{"reason": "...", "prompt_following": int, "consistency": int}}

The source images for our constructed multi-turn image editing benchmark, MSE-Bench, are sampled from MS-COCO (Lin et al., 2014) and LAION-Aesthetics (Schuhmann et al., 2022). Specifically, we randomly sample 6,000 images from each dataset and employ GPT-4o to perform prompt imagination, guided by criteria such as editing reasonability, aesthetics, consistency, and coherence. To facilitate this, we define a set of editing operations (e.g., add, remove, replace) and design a series of rules to instruct GPT-4o to simulate realistic and coherent multi-turn editing prompts from real users' perspectives. The instruction used in this process is illustrated above. Following prompt generation, we conduct careful human filtering to remove low-quality cases, resulting in a final set of 100 high-quality, category-balanced examples that constitute MSE-Bench. Additional examples are shown in Fig.13.

## C.7 SUPERVISED FINE-TUNING

After training on the constructed interleaved data from native videos, *i.e.*, VINCIE-10M, we carry out supervised fine-tuning to align the model with downstream editing tasks. Specifically, all of our SFT data comes from open-sourced datasets, including:

- OmniEdit-Filtered-1.2M[2] (Wei et al., 2024),

- Web-Image-3 and GRIT-Entity-New splits from X2I-subject-driven[3] proposed in Omni-Gen (Xiao et al., 2024),

- X2I2-video-editing, X2I2-inpaint-editing, X2I2-in-context-generation, and X2I2-in-context-editing splits from X2I2[4] proposed in OmniGen (Wu et al., 2025b),

- SEED-Data-Edit-Part3[5] proposed by SEED-X (Ge et al., 2024).

---

[2] https://huggingface.co/datasets/TIGER-Lab/OmniEdit-Filtered-1.2M
[3] https://huggingface.co/datasets/yzwang/X2I-subject-driven
[4] https://huggingface.co/datasets/OmniGen2/X2I2
[5] https://huggingface.co/datasets/AILab-CVC/SEED-Data-Edit-Part2-32

Table 6: Human evaluation on MSE-Bench based on editing success rate. * indicates use of context. Entries by gray denote proprietary models.

| Method | Human Evaluation | | | | |
|---|---|---|---|---|---|
| | Turn-1 | Turn-2 | Turn-3 | Turn-4 | Turn-5 |
| HQEdit (Hui et al., 2024) | 0.170 | 0.073 | 0.020 | 0.003 | 0.000 |
| UltraEdit (Zhao et al., 2024) | 0.310 | 0.062 | 0.015 | 0.002 | 0.000 |
| OmniGen (Xiao et al., 2024) | 0.333 | 0.035 | 0.002 | 0.000 | 0.000 |
| GPT-4o* | 0.872 | 0.783 | 0.755 | 0.642 | 0.491 |
| Ours* | 0.661 | 0.500 | 0.323 | 0.209 | 0.070 |

# D  ADDITIONAL EXPERIMENTAL RESULTS

## D.1  HUMAN EVALUATION ON MULTI-TURN IMAGE EDITING

To further verify the effectiveness and superiority of the proposed method for multi-turn image editing, we conduct human evaluations to assess editing success rates. The results are reported in Tab. 6. These findings validate the benefits of training on native video data, combined with supervised fine-tuning on pairwise editing examples, in enhancing multi-turn editing performance.

## D.2  CORRELATION BETWEEN GPT-4O AND HUMAN EVALUATION

Table 7: Correlation between automatic metrics and human evaluation

| Metric | GPT-4o vs Human | CLIP-T vs Human | CLIP-I vs Human |
|---|---|---|---|
| Pearson $r$ | **0.4858** ($p = 0.0000$) | 0.0817 ($p = 0.4191$) | -0.0549 ($p = 0.5875$) |
| Spearman $\rho$ | **0.4644** ($p = 0.0000$) | 0.0692 ($p = 0.4941$) | -0.0217 ($p = 0.8303$) |
| Kendall $\tau$ | **0.4154** ($p = 0.0000$) | 0.0502 ($p = 0.4963$) | -0.0195 ($p = 0.7921$) |

In our experiments (Sec.4), we primarily report GPT-4o evaluated success rates to assess multi-turn image editing performance. To validate the reliability of GPT-4o-based evaluation, we compute the correlation between GPT-4o scores and human judgments. As shown in Tab.7, we also compare other metrics such as CLIP-T and CLIP-I. The results demonstrate that GPT-4o correlates well with human evaluation, supporting its use as a reliable proxy for scoring multi-turn image editing.

## D.3  HUMAN EVALUATION FOR VLM ANNOTATION

To further verify the validity of the proposed automatic interleaved data construction pipeline, we conducted a human evaluation for VLM annotation on accuracy and recall, as shown in Tab. 8. While current VLMs are imperfect, they offer a scalable data annotation solution, achieving a favorable balance between quality and scalability. Similar to prior works (Brooks et al., 2023), our work aims to explore continual pre-training on the constructed large-scale interleaved corpus, where data scale is critical and minor annotation noise is tolerable. Besides, we believe the proposed automatic data construction pipeline will become increasingly effective, with the rapid advancement of large multimodal models.

Table 8: Human evaluation for VLM annotation on 500 data instances randomly sampled from our training dataset.

| Metric | Score |
|---|---|
| Accuracy | 75.14% |
| Recall | 69.06% |

## D.4  ADDITIONAL ABLATION STUDY

**Impact of RoPE and Attention**. Based on a video foundation model, VINCIE continues pre-training on the constructed interleaved text-image data. In the foundation model, we first concatenate text tokens and video tokens into a sequence, perform Rotary Position Embedding (RoPE) (Su et al., 2024) on it, and then feed it to multiple MM-DiT (Esser et al., 2024) layers for video modeling. Full bidirectional attention is adopted in each layer for thorough intra-modal and cross-modal interaction.

Table 9: Impact of segmentation (seg.) prediction and camera prompt engineering (PE), *i.e.*, inserting "[###CAMERA: None###]" before each user prompt, on consistency evaluated by CLIP-I and DINO scores on Magicbrush (Zhang et al., 2023a).

| Train | Inference | CLIP-I Score | | | DINO Score | | |
|---|---|---|---|---|---|---|---|
| | | Turn-1 | Turn-2 | Turn-3 | Turn-1 | Turn-2 | Turn-3 |
| w/o Seg. | - | 0.875 | 0.824 | 0.784 | 0.765 | 0.663 | 0.592 |
| w/ Seg. | - | 0.880 | **0.832** | 0.797 | 0.786 | 0.680 | 0.604 |
| w/ Seg. | Camera PE | **0.884** | **0.832** | **0.798** | **0.798** | **0.681** | **0.612** |

Table 10: Ablation study on MSE-Bench (GPT-4o evaluated success rate), to assess the impact of RoPE and Attention.

| RoPE | Attention | Turn-1 | Turn-2 | Turn-3 | Turn-4 | Turn-5 |
|---|---|---|---|---|---|---|
| text-then-image | full | **0.968** | **0.360** | **0.320** | 0.238 | 0.160 |
| interleaved | full | 0.933 | 0.338 | 0.308 | **0.245** | **0.183** |
| interleaved | block-causal | 0.880 | 0.290 | 0.230 | 0.200 | 0.120 |

To explore the impact of RoPE and Attention, we design three variants and conduct a performance comparison on MSE-Bench, as shown in Tab. 10. Considering the foundation model has carried out large-scale pre-training on text-video data, it has attained strong prior knowledge based on text-then-image RoPE and full attention. This strategy achieves the best performance on Turn-1 to Turn-3. However, the interleaved RoPE gradually outperforms it as the sequence length increases. One reason is that the interleaved RoPE arranges the text and image more naturally than the trivial text-then-image strategy. Finally, block-causal attention performs the worst, which may be attributed to the limited modality interaction. However, block-causal attention shows strong potential, offering flexibility in next-block modeling, support for prefill decoding to enable efficient inference, and compatibility with LLMs, which we leave for future work.

**Impact of Camera Motion in Training Video Data**. In most editing scenarios, consistency is highly required. To delve into possible entanglement issues of camera and object movement, we have adopted a disentanglement learning strategy consisting of: 1) Explicit annotation of camera change (see the instruction in Sec. C.2); 2) Incorporation of camera prompt wrapped in special tokens, such as "[###CAMERA: pan-left###]", during training; 3) Use of static camera prompt, *i.e.*, "[###CAMERA: None###]", during inference. This strategy enables the model to disentangle camera movement from object dynamics, allowing flexible camera control based on application needs. The results shown in Tab. 9 verify its effectiveness in improving consistency.

## D.5    ADDITIONAL PERFORMANCE COMPARISON ON STORY KEYFRAME GENERATION

Tab. 11 provides a quantitative comparison of story keyframe generation performance between our method and the recent method, *i.e.*, In-context LoRA Huang et al. (2024) on the benchmark introduced in LCT (Guo et al., 2025), serving as empirical evidence of the effectiveness of our approach. In this work, we aim to introduce a general video-driven learning framework to unlock in-context image editing and generation, with story keyframe generation being one potential application. Due to the limited time, we focus on multi-turn image editing, while more comprehensive evaluation of other capabilities, including story generation, is left for future work.

Table 11: Our method vs. In-context LoRA (Huang et al., 2024) with human evaluation on the benchmark introduced in LCT (Guo et al., 2025) for story keyframe generation. Evaluation is carried out from two aspects: prompt following and consistency.

| Metric | Win | Fail | Tie |
|---|---|---|---|
| Prompt Following | 55.10% | 16.30% | 28.60% |
| Consistency | 43.80% | 2.10% | 54.20% |

Change the blender's exterior to have a metallic finish.

Replace the wooden table with a white marble surface and add another small red decoration.

Change the grass court to a clay court.

It appears to be snowing.

Add a rainbow, and blue sky and cloudy days interweave.

Transform it into an oil painting style.

Generate a monochrome-style animation.

Transform this image into a Pointillist artwork.

Figure 14: **Zero-shot** qualitative results of single-turn image editing on cases uncommonly present in video data. The model was only trained with interleaved session data from video, T2I data, and T2V data.

# E  ADDITIONAL APPLICATION EXAMPLES

## E.1  IMAGE EDITING

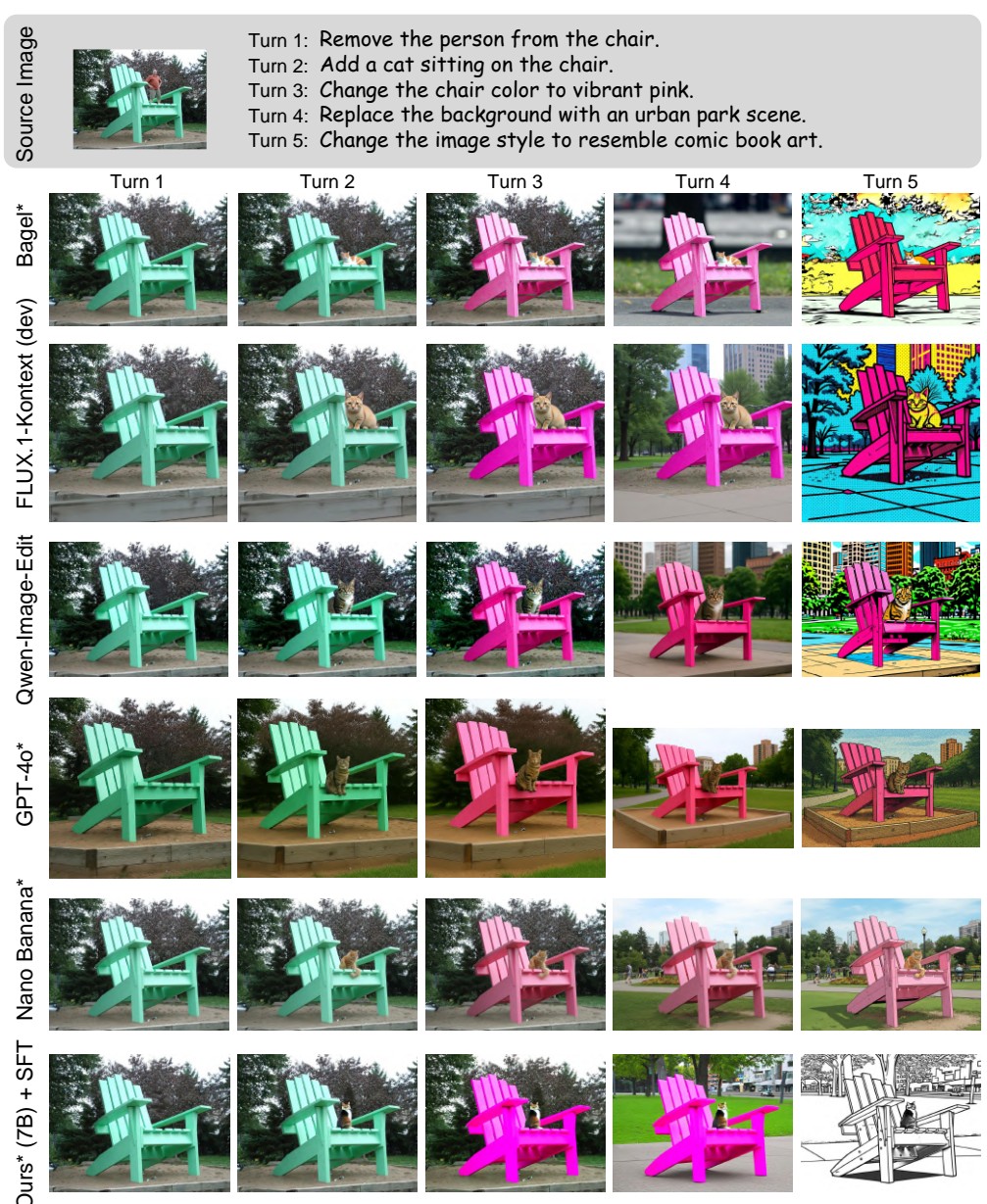

Figure 15: Qualitative comparison (1/4) between our method and recent baselines on MSE-Bench.

**Single-turn Image Editing.** In addition to common scene changes present in video data, we observe that our model generalizes well to uncommon cases, such as abrupt environmental shifts, complex style transfers, and material transformations (Fig. 14). This capability may arise for two reasons. First, although infrequent, such patterns (*e.g.*, environmental changes) are still present in our training corpus. Second, the model is initialized from a video foundation model that has been extensively pre-trained on both T2I and T2V data, enabling it to internalize high-level concepts such as style and material. These derived capabilities can be naturally transferred to the image editing setting.

**Multi-turn Image Editing.** As shown in Fig. 15, we compare our method with several baselines, including HQ-Edit (Hui et al., 2024), UltraEdit (Zhao et al., 2024), OmniGen (Xiao et al., 2024),

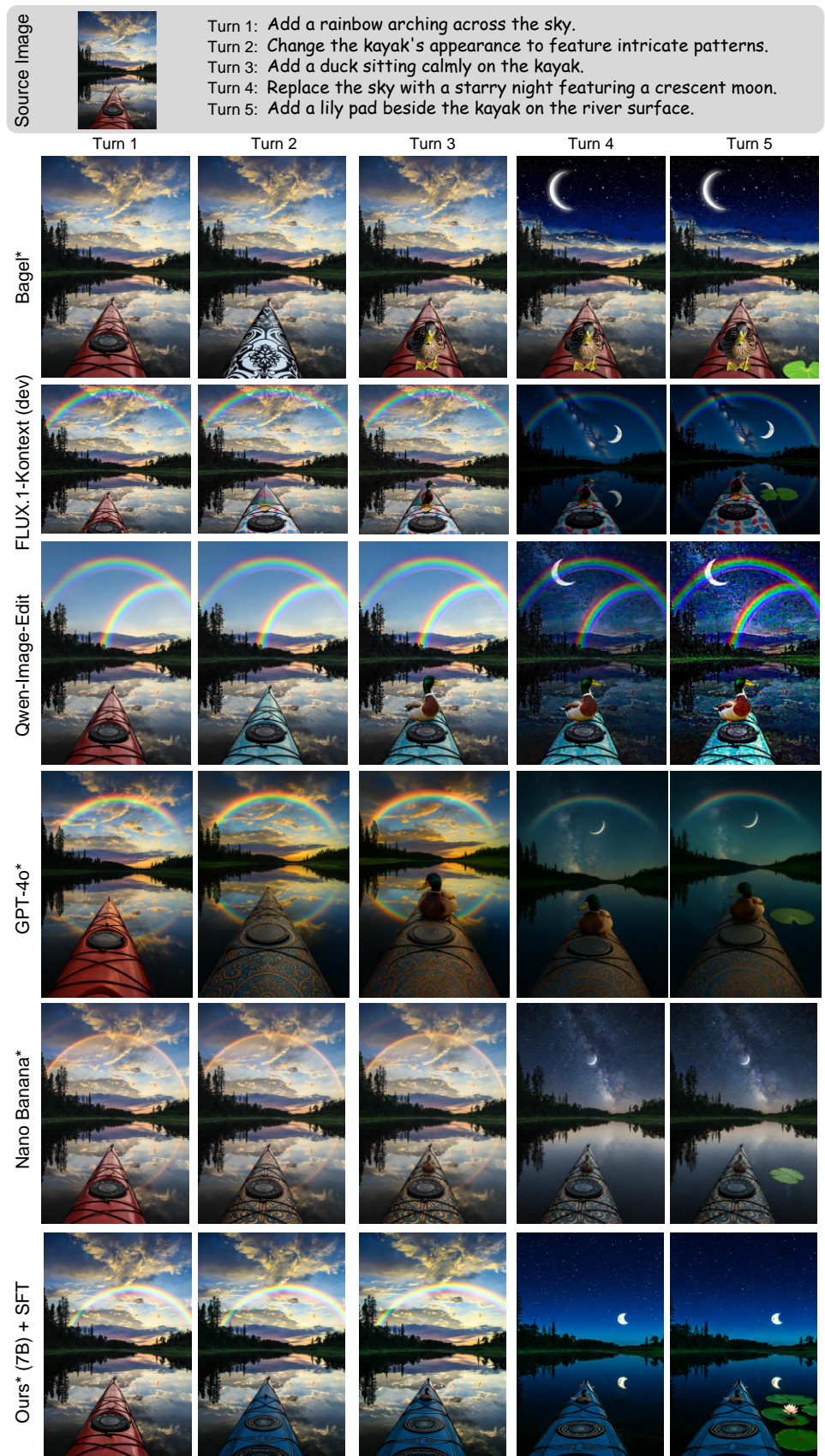

Figure 16: Qualitative comparison (2/4) between our method and recent baselines on MSE-Bench.

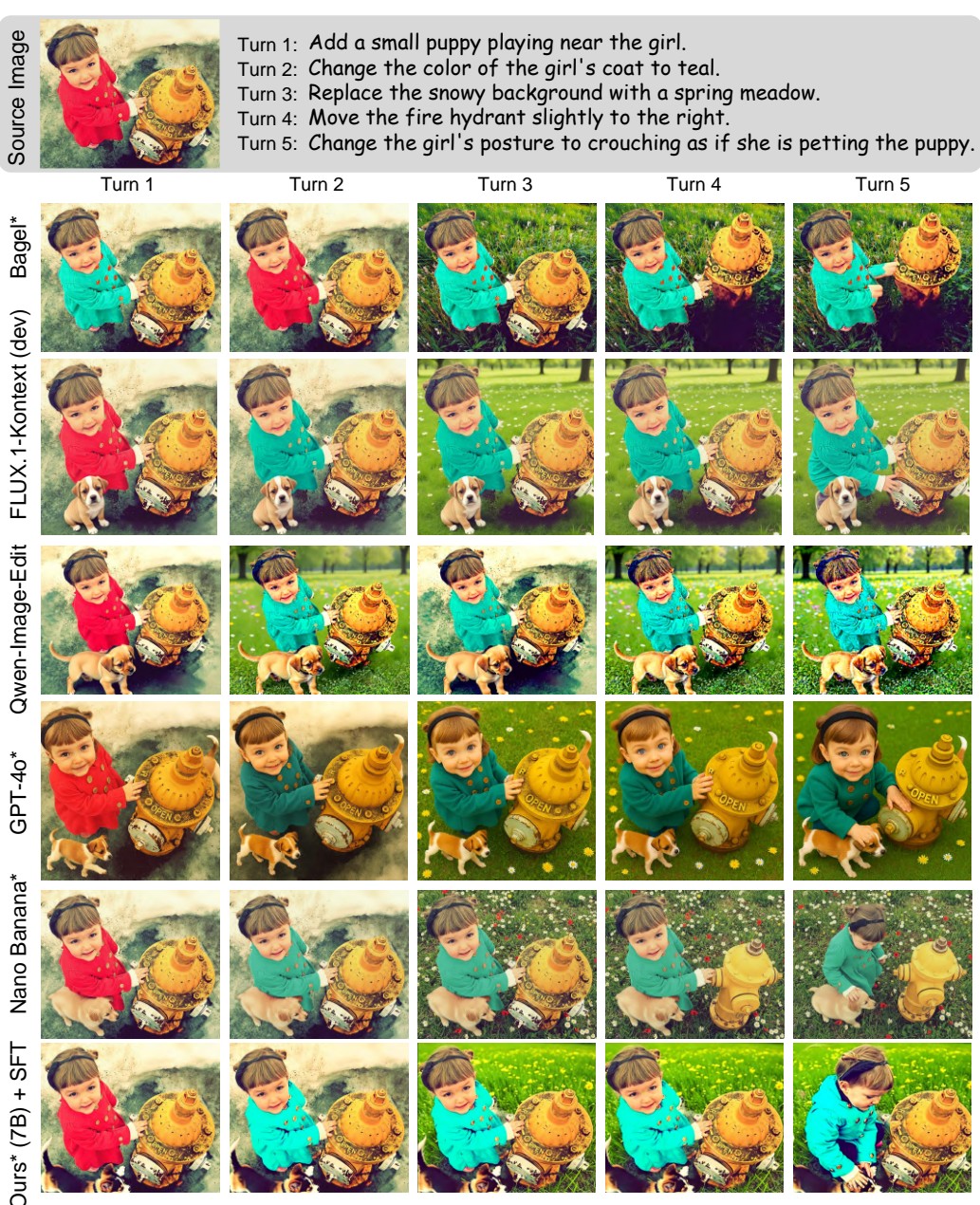

Figure 17: Qualitative comparison (3/4) between our method and recent baselines on MSE-Bench.

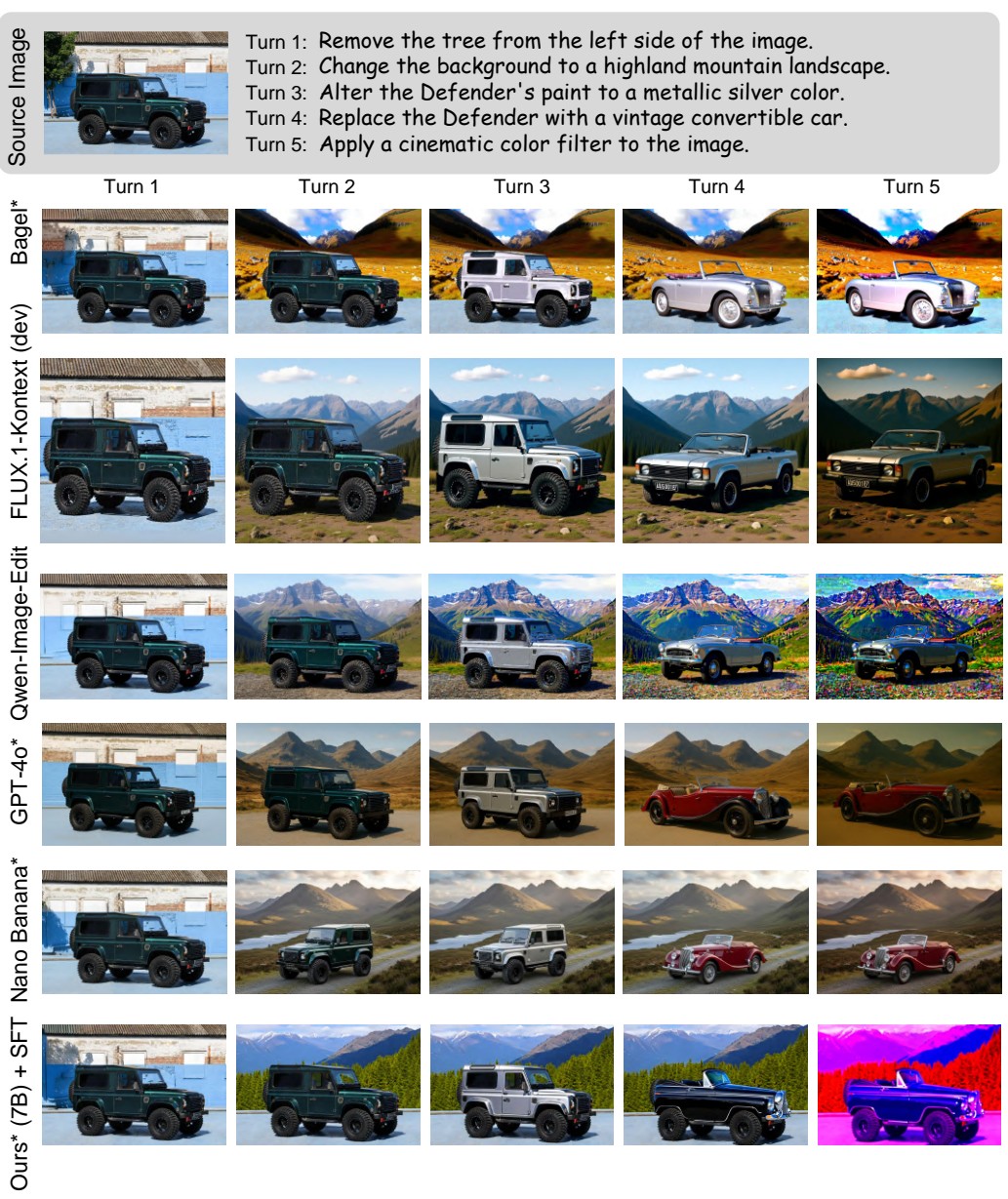

Figure 18: Qualitative comparison (4/4) between our method and recent baselines on MSE-Bench.

and GPT-4o. The results reveal several key observations: 1) Most existing models suffer from error accumulation, leading to increasingly severe artifacts across editing turns. 2) These accumulated errors often degrade prompt-following performance, where the model fails to execute edits as instructed once artifacts dominate. 3) While GPT-4o—a strong proprietary model—achieves competitive results, it may exhibit inconsistencies in some cases compared to our method. 4) Overall, these comparisons highlight the effectiveness of training on native video data for achieving coherent and prompt-aligned multi-turn image editing. Additional qualitative examples are provided in Fig. 26, Fig. 27, Fig. 28, and Fig. 29, further demonstrating the strong prompt-following and consistency of our approach across multiple editing turns.

## E.2  MULTI-CONCEPT COMPOSITION

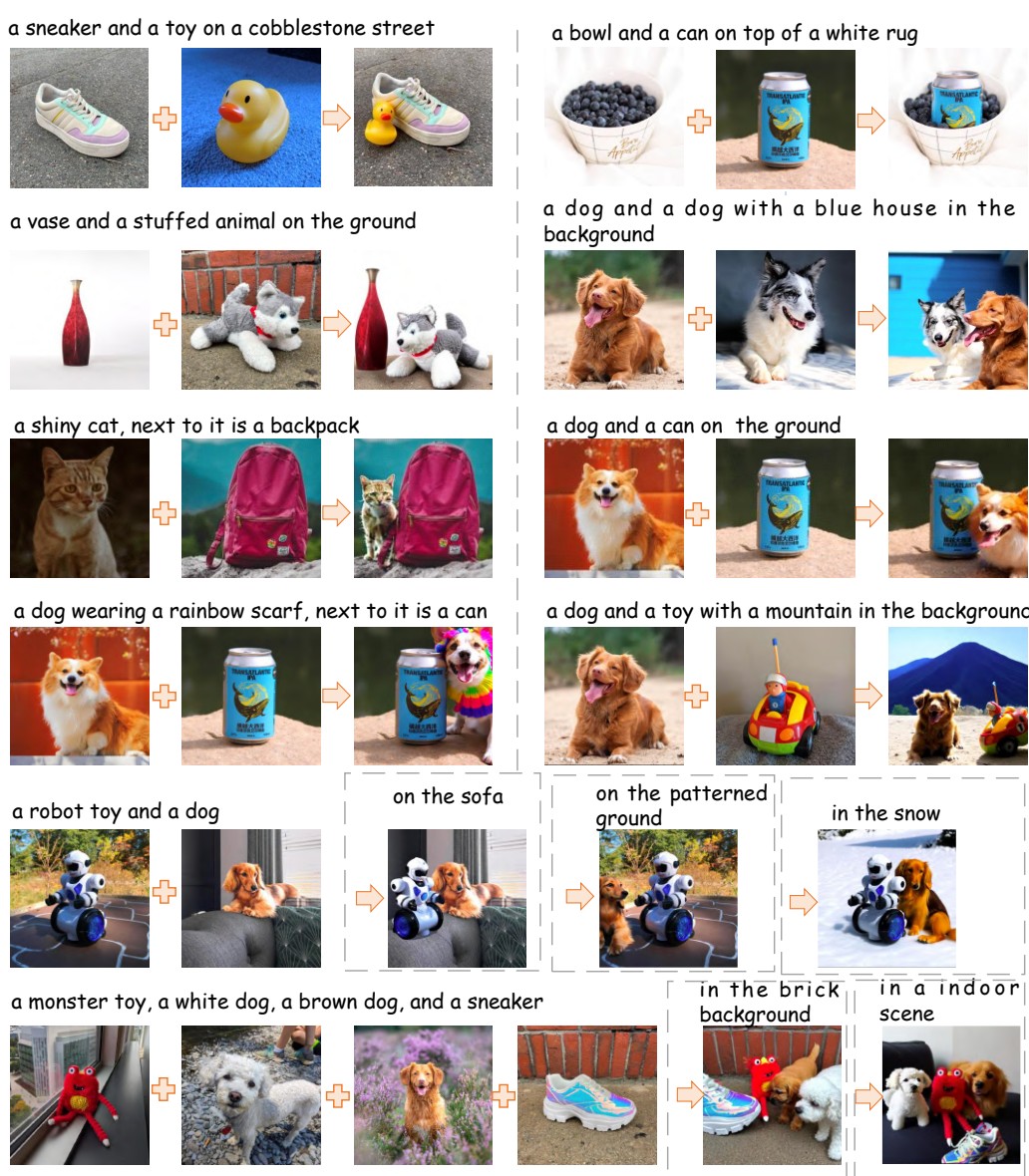

Figure 19: **Zero-shot** qualitative results of multi-concept composition (in-context generation) achieved by our method (without any fine-tuning).

**In-context Image Generation**.  In Fig. 19, we present qualitative results on in-context image generation for multi-concept composition, which requires both composition and strong identity

Replace the woman in <IMG2> with the man in <IMG1>.

Add the bowl in <IMG1> in front of the toy bear held by the old woman in <IMG2>.

Apply the background color from <IMG1> to <IMG2>.

Swap the background in <IMG2> with the same red brick background from <IMG1>.

Make the girl in <IMG2> have the same expression with the woman in <IMG1>.

Recolor the helmet in <IMG2> using the color of the sweater from <IMG1>.

Figure 20: Qualitative results of multi-concept composition (in-context editing) achieved by our method. Our model is fine-tuned on the X2I2 (Wu et al., 2025b) dataset, but the shown transferred concepts (*e.g.*, background, color, and expression) are uncommon in X2I2. It demonstrates the strong generalization ability conferred by pre-training on video-based interleaved data.

preservation. These examples demonstrate that only training on video data (without any fine-tuning) can effectively unlock compositional capabilities, despite the rarity of such patterns in typical video content. This emergent behavior highlights the potential of video-based pre-training. Further scaling of model capacity, compute resources, and video data may enable the emergence of even more advanced capabilities. For instance, enhanced identity preservation enables more effective personalization (Wang et al., 2025; Zhao et al., 2025; 2026).

**In-context Image Editing**. In addition, we conducted further supervised fine-tuning on the X2I2 (Wu et al., 2025b) dataset to explore more advanced multi-concept composition abilities. The qualitative results (Fig. 20) on in-context image editing indicate that even lightweight SFT substantially incentivizes more powerful compositional editing abilities, highlighting the effectiveness of our video-driven pre-training. Notably, compared with Fig. 19, **our fine-tuned model generalizes beyond object-centric concepts, such as background, color, and expression**, despite these concepts being uncommon in the fine-tuning dataset (Wu et al., 2025b).

### E.3 STORY GENERATION

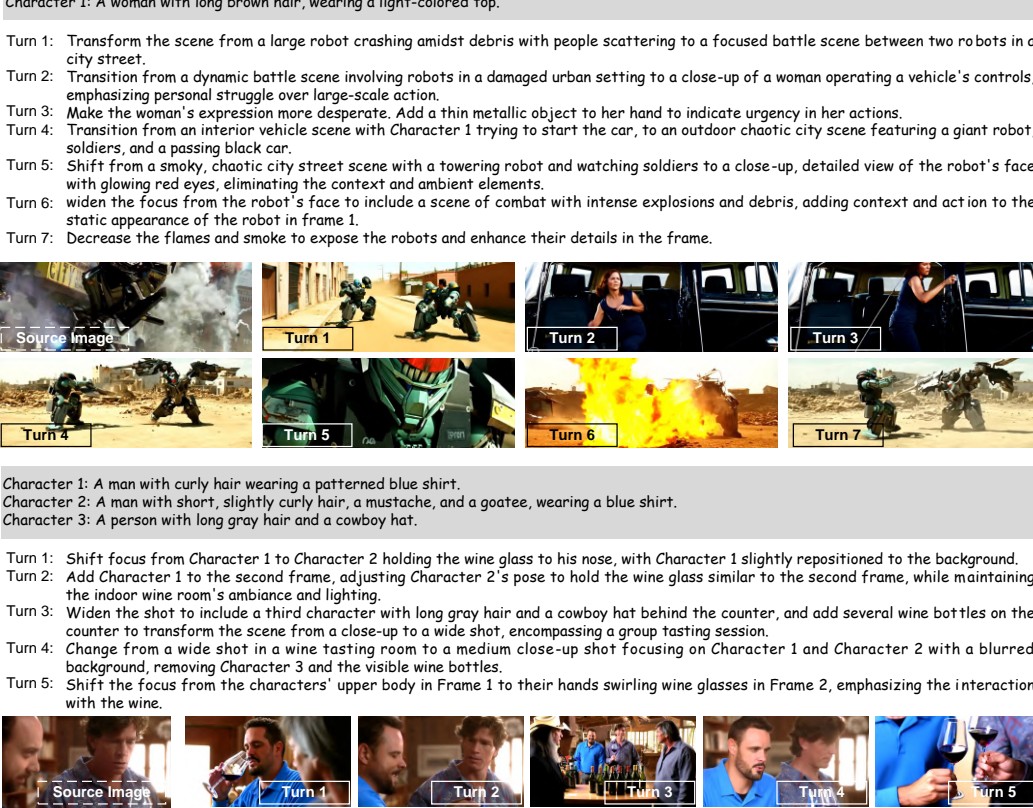

Figure 21: More qualitative results of story generation achieved by our method.

Since our method is trained on native video data, it inherently captures the underlying storylines present in the sequences. As illustrated in Fig.21, we formulate story generation as a multi-turn image editing task, guided by transition prompts between key frames during inference. These examples showcase the model's ability to follow prompts while maintaining coherence and consistency across turns. When combined with existing long video generation methods(Guo et al., 2025), our approach has the potential to enhance top-down planning for generating coherent long-form story videos.

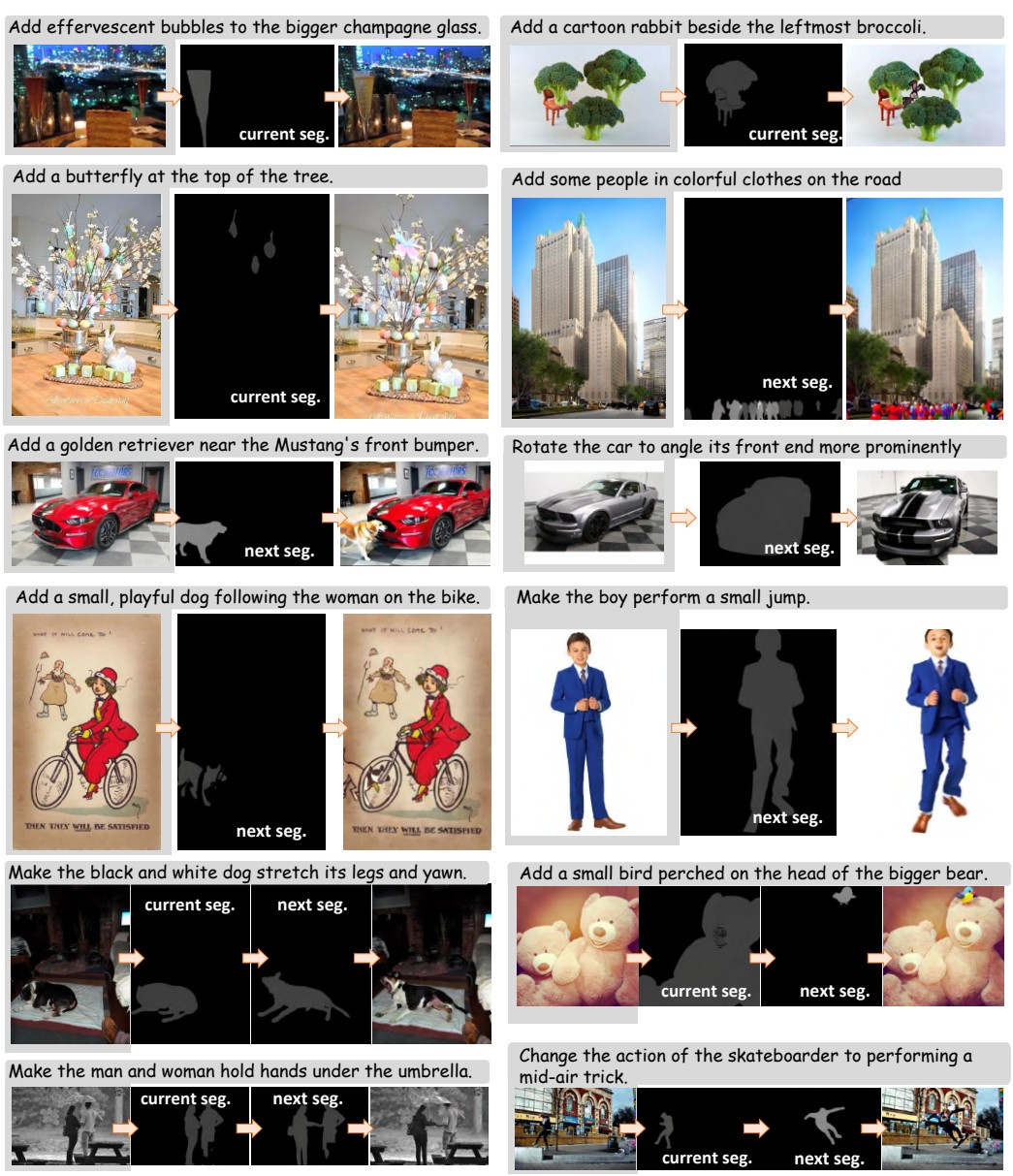

Figure 22: More qualitative results of Chain-of-Editing.

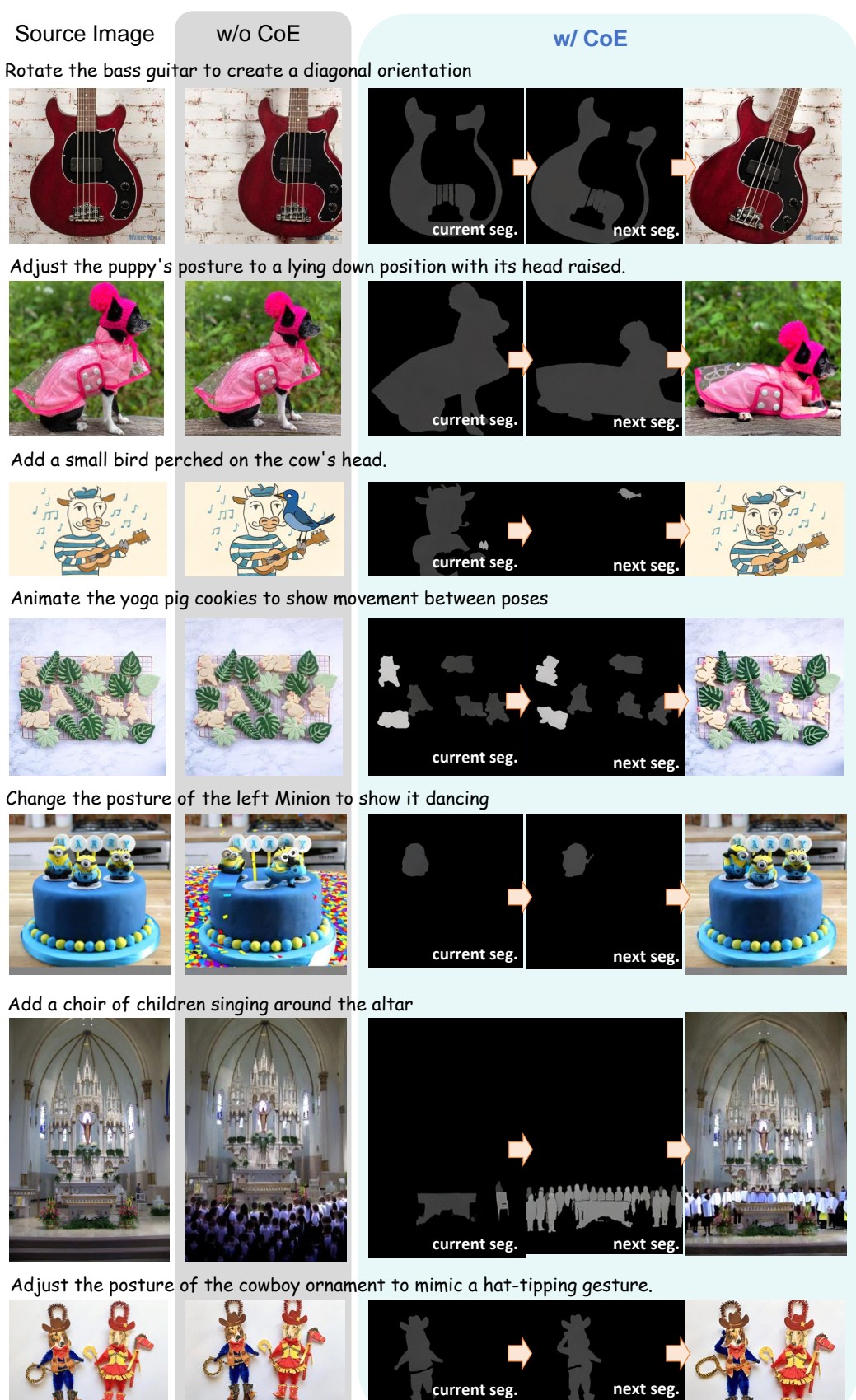

Figure 23: Qualitative comparison between w/o Chain-of-Editing (CoE) and w/ CoE.

### E.4 CHAIN-OF-EDITING

In Tab. 3, we show the effectiveness of chain-of-editing, *i.e.*, predicting segmentation maps before performing image editing. The predicted segmentation maps could be viewed as a kind of "thoughts". In Fig. 22, we show more qualitative results for challenging cases to demonstrate the effectiveness of CoE.

### E.5 DRAG-BASED IMAGE EDITING

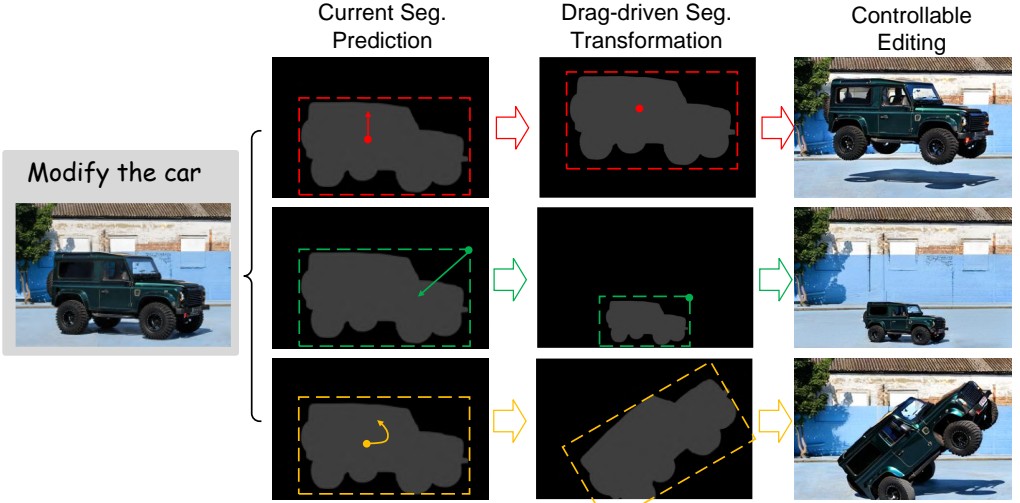

Figure 24: Qualitative results of drag-based image editing.

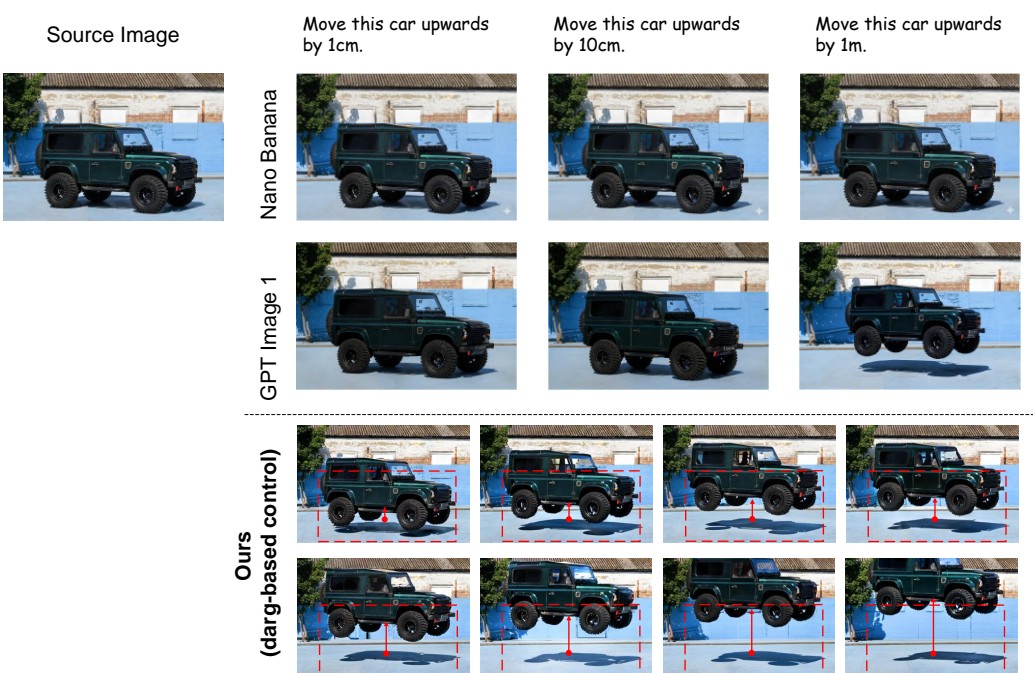

Figure 25: Qualitative comparison for subtle displacement editing.

The current and next segmentation prediction tasks introduced in Sec. 3.3 not only support progressive planning and generation, but also enable controllable editing for enhanced user interaction. One such

application is drag-based image editing for object displacement, scaling, and rotation, as illustrated in Fig.24. In this setting, users first provide an editing prompt to localize the RoE. Then, drag operations are applied to perform geometric transformations of the RoE. The transformed segmentation map driven by the transformation is incorporated into the context, allowing the model to generate a target image that adheres to the specified edits.

Despite the strong understanding capabilities (Liu et al., 2018; Qu et al., 2024a) of VLMs, they may still struggle to detect subtle semantic or visual differences when two frames differ only minimally. As illustrated in Fig. 25, we first present qualitative results from the most advanced proprietary systems—GPT Image 1 (OpenAI, 2025a) and Nano Banana (DeepMind & Gemini, 2025)—on the task of subtle displacement editing. We then showcase our drag-based editing results, demonstrating that this challenging requirement can be effectively addressed through the more flexible and fine-grained control (*i.e.*, drag). This comparison highlights the versatility of our method.

## F    LIMITATIONS

**Discussion of Other Potential Limitations**. First, we use T5 to encode text, which restricts the model's ability to comprehend complex instructions and generate nuanced textual outputs. Integrating a vision-language model (VLM) into the framework could significantly improve this capability. Second, while our framework demonstrates preliminary but promising emerging abilities, these can be further enhanced through supervised fine-tuning (SFT) on high-quality, application-specific datasets. Lastly, due to the high cost of querying VLM, we annotated only 10M training samples. Expanding both the model size and the dataset scale presents an exciting avenue for future research.

## G    FUTURE WORK

In the future, we aim to solve more challenging image creation tasks (Qu et al., 2023; Yang et al., 2024a; Qu et al., 2024b) with complex and compositional prompts, by exploring multimodal chain-of-thought. Besides, post-training (Qu et al., 2025b; Zhou et al., 2025c; Gong et al., 2025) would stimulate more potential interesting abilities endowed by learning from videos. Finally, by introducing retrieved images (Qu et al., 2025a; Chen et al., 2022; Qu et al., 2021; Wen et al., 2024) into context, our model could achieve knowledge-intensive visual creation scenarios via retrieval-augmented generation.

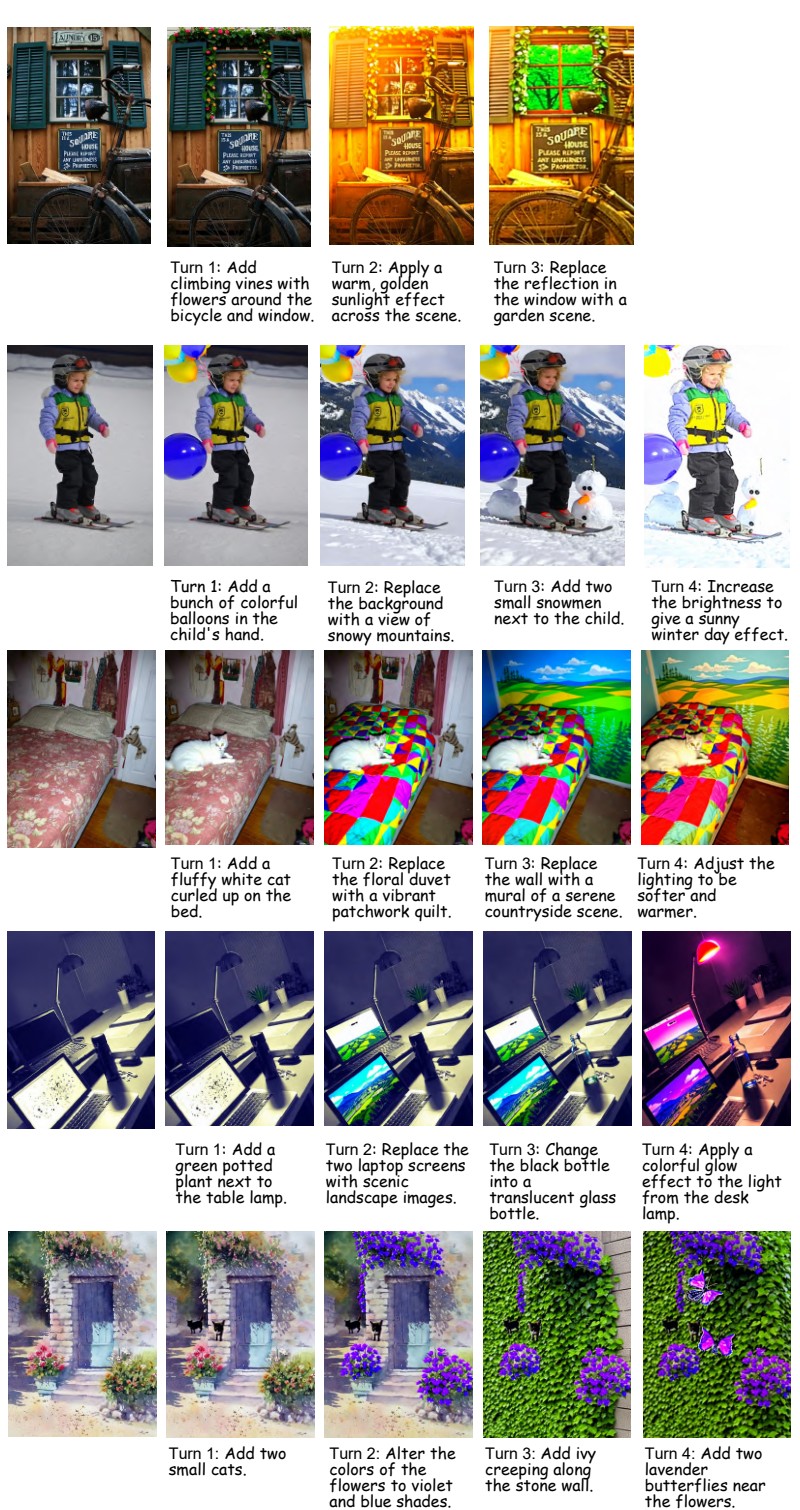

Figure 26: More qualitative results (1/4) of multi-turn image editing achieved by our method.

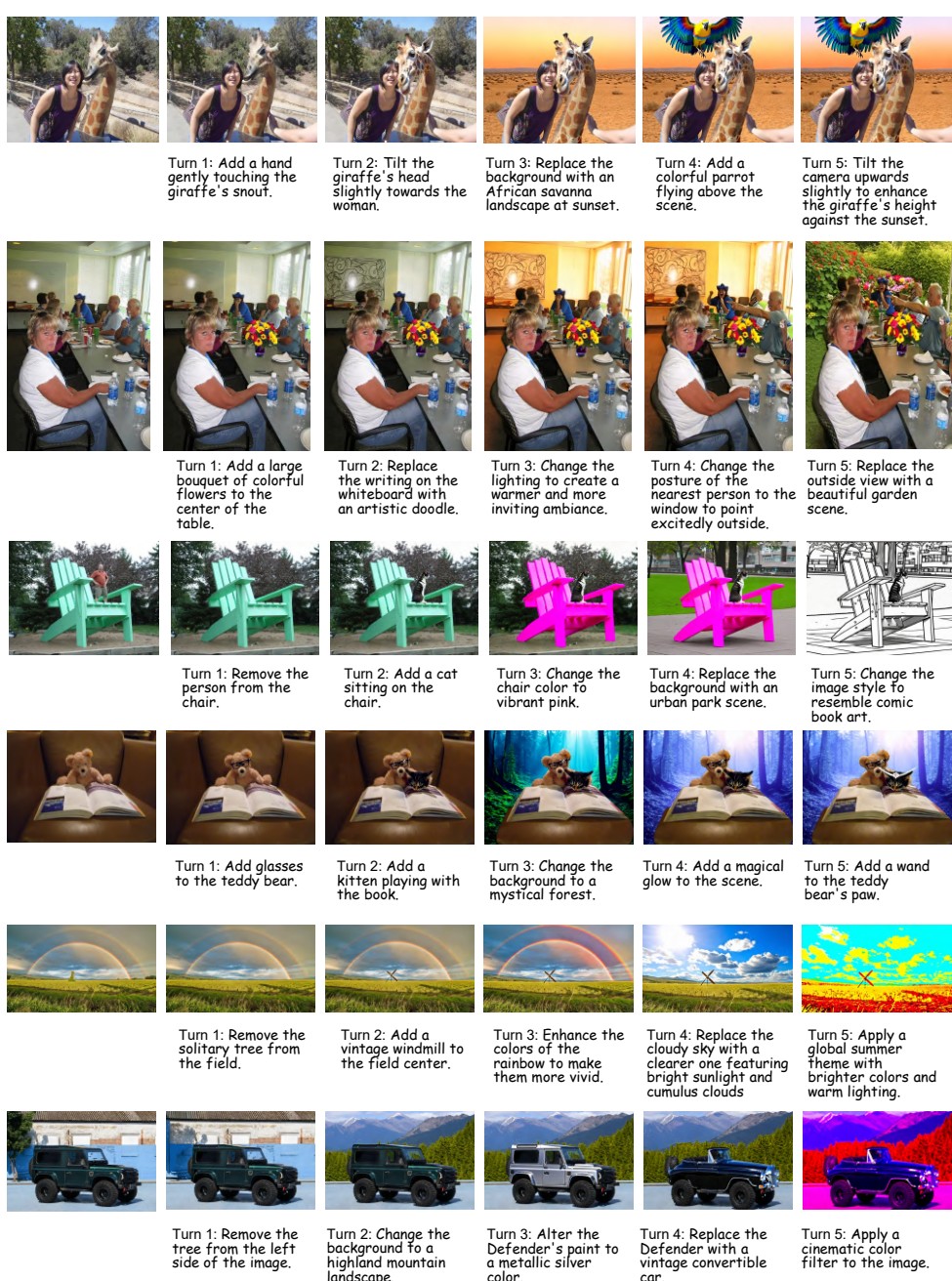

Figure 27: More qualitative results (2/4) of multi-turn image editing achieved by our method.

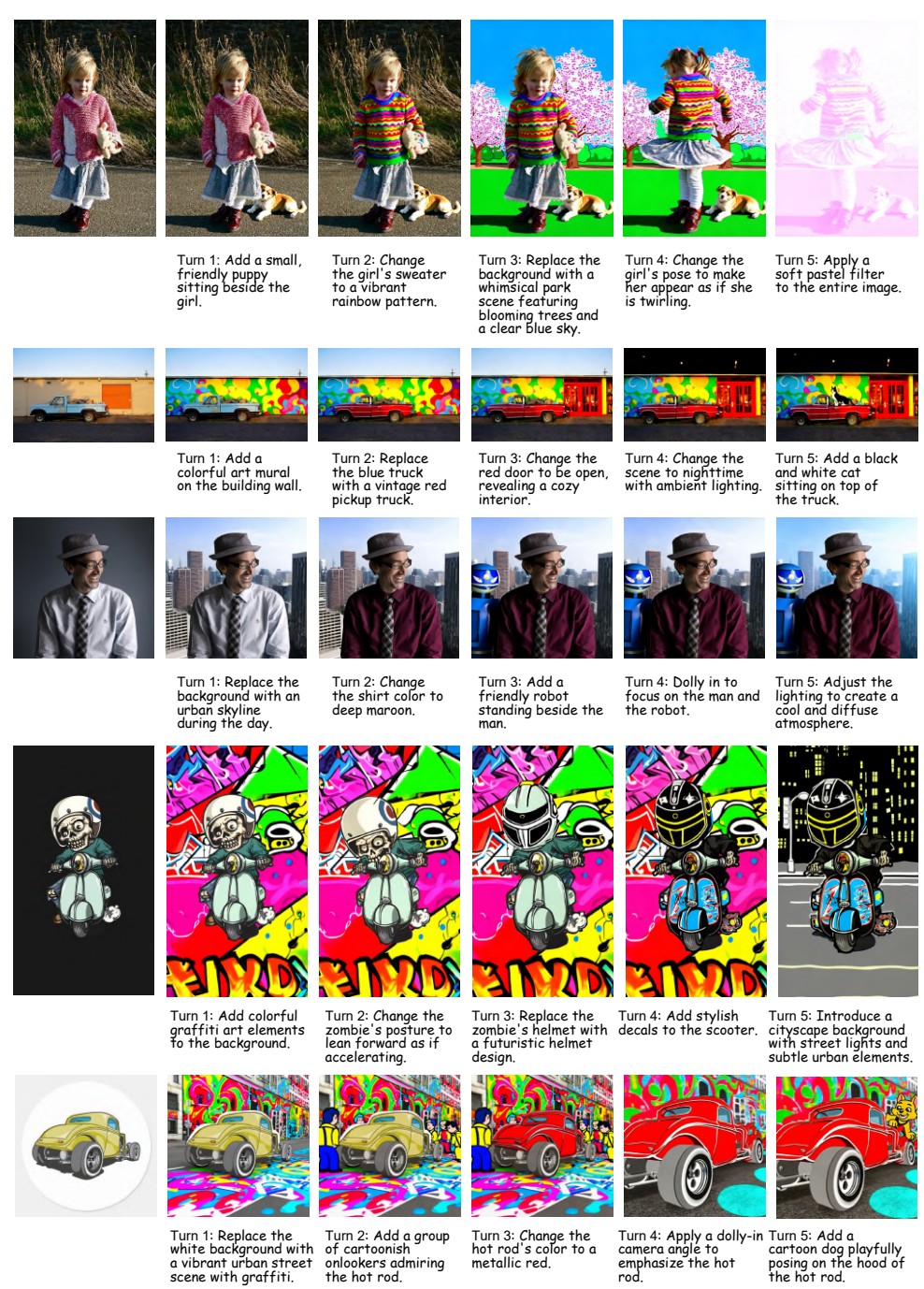

Figure 28: More qualitative results (3/4) of multi-turn image editing achieved by our method.

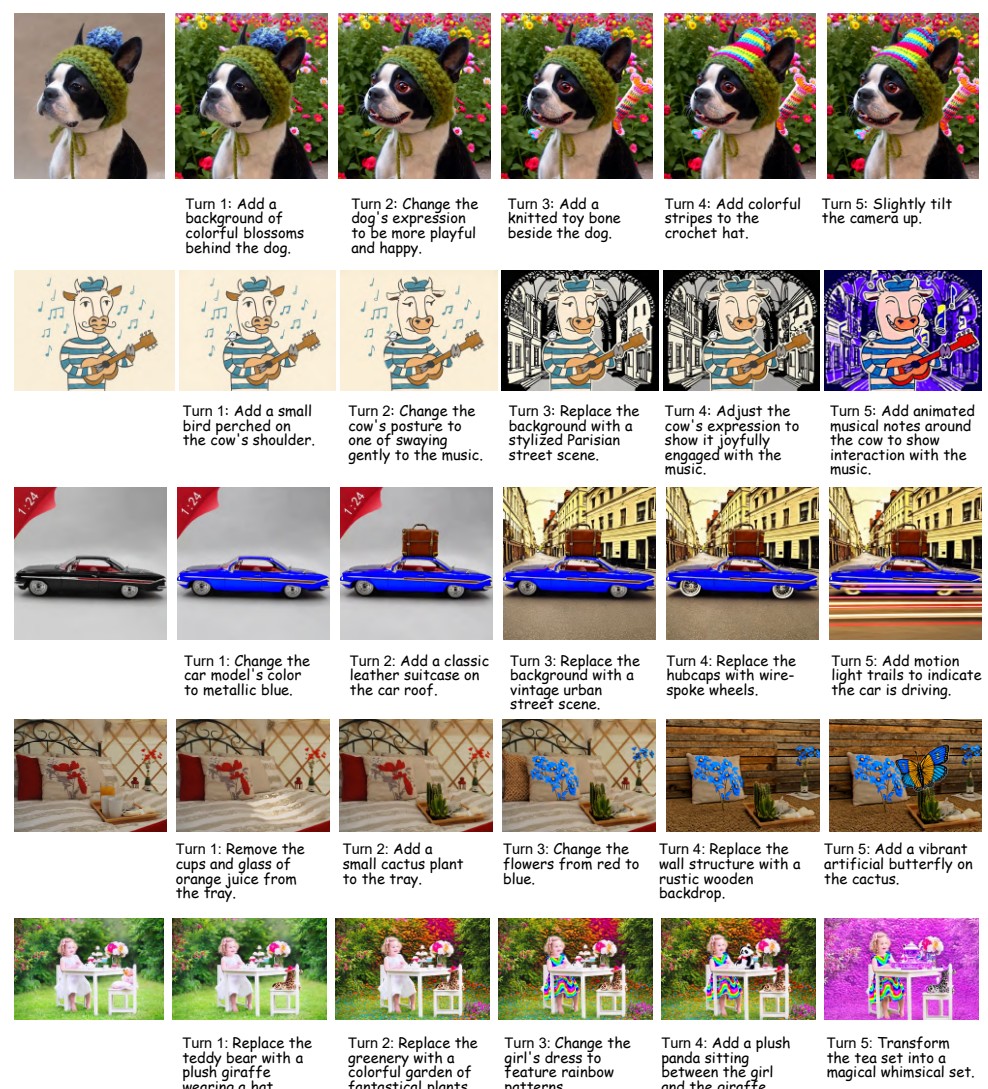

Figure 29: More qualitative results (4/4) of multi-turn image editing achieved by our method.

