# OpenReview forum: "VINCIE: Unlocking In-context Image Editing from Video"
_ICLR.cc/2026/Conference — ICLR 2026 Poster_

### Official Review · Reviewer_6Gue · 2025-10-29

**Soundness:** 3
**Presentation:** 2
**Contribution:** 3
**Rating:** 4
**Confidence:** 4

**Summary:**

The paper proposes VINCIE, a video-driven approach to in-context image editing that learns multi-turn image editing directly from videos rather than paired images. In the method, videos are converted into interleaved multimodal sequences of sparsely sampled frames, VLM-written transition texts, and segmentation masks for regions of editing. The method leverages a DiT with block-wise causal attention trained on three proxy tasks: next-image prediction, current-segmentation prediction, and next-segmentation prediction. The authors also introduce MSE-Bench, a 5-turn editing benchmark, and report strong results on MSE-Bench and MagicBrush, with additional analyses on context, segmentation prediction, and data scaling.

**Strengths:**

1. The proposed pipeline of training multi-turn image editing models from video data is novel and intuitively effective.
2. The proposed MSE-Bench can benefit the image editing community and more properly evaluate multi-turn image editing models.
3. VINCIE achieved strong multi-turn image editing performance compared to previous art.

**Weaknesses:**

1. What are the training objectives for the segmentation prediction tasks NSP and CSP? Are they MSE or some other losses between the predicted and GT segmentation masks?
2. Can the author provide some intuition on why adding the next segmentation prediction task during inference benefits MagicBruch scores but damages MSE-Bench scores (compared to CS->I)?
3. From Table 4, it seems that the Dummy context is more effective than providing the editing history to the model. Is this the correct interpretation of this table? Does that mean the Dummy context itself is enough for multi-turn image editing?
4. Based on Section 4.4, the camera movement and subject position shift problems can be solved by incorporating the segmentation prediction task during inference. Can the author elaborate more on why this prediction task helps stabilize the movement of the subjects in the image? Is it because the predicted segmentation mask always tends to be the same mask as in the previous editing round?
5. The paper mentioned both full attention and block-wise causal attention in the method section. Which one is the method that is used in the final version of VINCIE? From Section 1, it seems that block-wise causal attention is actually used in VINCIE. If this is the case, what is the point of mentioning the full attention design? Or does the model support both attention mechanisms (across different editing turns) natively?
6. I would suggest adding some description on how the model is used during inference, especially on the different modes that the model can be used, since the model accepts a wide variety of input conditions and these inputs can be quite flexible. Right now it is a bit confusing on how the model operates during inference: for example, when adding the segmentation mask prediction task during inference, will the predicted segmentation mask be used as part of the input condition to the model, or will it only be predicted, but not used during the next edited image prediction?

Overall, I find the method and results to be promising, but I believe the current paper's presentation can be improved. I would be happy to raise my score if the concerns can be addressed.

**Questions:**

See weaknesses above

---

> ### Author Response · Authors · 2025-11-21
> **Author Response to Reviewer 6Gue (1/3)**
>
> We are deeply grateful to the reviewer for the thorough evaluation and thoughtful feedback. Your comments are highly constructive and have been instrumental in helping us refine this manuscript. Our point-by-point responses are provided below.
>
> ---
>
> > (W1) What are the training objectives for the segmentation prediction tasks NSP and CSP? Are they MSE or some other losses between the predicted and GT segmentation masks?
>
> We unify the generation and segmentation prediction tasks under the denoising **MSE** training objective, to facilitate ability transfer between different modalities.
>
> ---
>
> > (W2) Can the author provide some intuition on why adding the next segmentation prediction task during inference benefits MagicBruch scores but damages MSE-Bench scores (compared to CS->I)?
>
> Thank you for the insightful comments. As shown in Table 3, we primarily focus on **consistency** performance on the MagicBrush dataset, where the CS→NS→I strategy achieves the best results. This improvement stems from the inclusion of more segmentation maps—current segmentation (CS) and next segmentation (NS)—which serve as stronger structural constraints, enhancing layout adherence. In essence, these additional maps act as *richer context or condition tokens* (approximately three times more than in pure image generation), thereby improving in-context learning.
>
> For MSE-Bench, we adopt a more comprehensive evaluation using editing success rate as the primary metric. Here, the CS→NS→I strategy slightly underperforms CS→I, which may be due to **task-specific advantages conferred by different strategies**.
> To better understand this, we report detailed performance—prompt following (PF), consistency (C), and overall (OV) —across five common editing categories in the table below. The best overall and PF performance for each task is highlighted in bold.
>
>
> | Category | CS→I |  |  | NS→I |  |  | CS→NS→I |  |  |
> |------|------|------|------|------|------|------|------|------|------|
> |      | OV   | PF   | C    | OV   | PF   | C    | OV   | PF   | C    |
> | add | 6.76 | 6.89 | 8.81 | **7.03** | **7.14** | 8.70 | 5.49 | 5.65 | 8.73 |
> | background | 4.38 | 4.81 | 6.11 | 4.43 | 4.65 | 7.30 | **5.30** | **5.62** | 7.86 |
> | color | 4.30 | 4.68 | 6.68 | **4.86** | **5.43** | 6.76 | 4.70 | 5.24 | 7.54 |
> | local | 4.92 | 5.22 | 7.70 | 4.81 | 4.97 | 8.38 | **5.76** | **5.84** | 8.22 |
> | remove | **6.46** | **6.84** | 8.59 | 5.11 | 5.30 | 8.14 | 5.86 | 6.30 | 8.24 |
> | all | 5.36 | 5.69 | 7.58 | 5.25 | 5.50 | 7.85 | **5.42** | **5.73** | 8.12 |
>
> We have the following observations and analysis:
> - CS→I excels in the remove category, likely because removal relies heavily on grounding (i.e., identifying the correct region using CS).
>
> - NS→I performs best on add and color, where layout planning is more critical (e.g., deciding where and how to add a new object).
>
> - For more complex edits like background and local, the step-wise nature of CS→NS→I yields better results by leveraging intermediate reasoning.
>
> - Notably, this experiment uses a uniformly sampled test set across all five categories, where the CS→NS→I chain-of-editing strategy demonstrates superior overall performance.

---

> ### Author Response · Authors · 2025-11-21
> **Author Response to Reviewer 6Gue (2/3)**
>
> > (W3) From Table 4, it seems that the Dummy context is more effective than providing the editing history to the model. Is this the correct interpretation of this table? Does that mean the Dummy context itself is enough for multi-turn image editing?
>
> Thank you for raising this valuable concern. As shown in Table 4, the Dummy Context variant outperforms the w/o Context baseline and even surpasses the variant using actual editing history. However, we observe a *slight degradation in prompt following performance (CLIP-T) as the number of editing turns increases*.
>
> To further clarify this trend, we conducted an extended evaluation on the proposed MSE-Bench, enabling a broader comparison across the three variants, as detailed below.
>
>
> | Method | Turn1 |      |      | Turn2 |      |      | Turn3 |      |      | Turn4 |      |      | Turn5 |      |      |
> |--------|--------|------|------|--------|------|------|--------|------|------|--------|------|------|--------|------|------|
> |        | PF     | SC   | OV   | PF     | SC   | OV   | PF     | SC   | OV   | PF     | SC   | OV   | PF     | SC   | OV   |
> | w/o Context | 8.71 | 8.83 | 8.65 | 6.85 | 7.56 | 6.96 | 5.09 | 5.61 | 5.10 | 3.79 | 3.91 | 3.74 | 2.25 | 3.11 | 2.44 |
> | w/ Dummy Context | **8.88** | **9.18** | **8.88** | 7.52 | **7.51** | **7.35** | 5.88 | 6.05 | 5.80 | 4.93 | 4.91 | 4.82 | 3.51 | 4.08 | 3.61 |
> | w/ History Context | 8.65 | 8.85 | 8.62 | **7.53** | 7.21 | 7.26 | **6.15** | **6.09** | **5.97** | **5.30** | **5.04** | **5.10** | **3.85** | **4.41** | **3.94** |
>
>
>
> The results indicate that while Dummy Context outperforms w/o Context, its performance declines relative to real history context as the number of turns increases. This suggests that the gains from Dummy Context stem primarily from the increased number of condition tokens, which help improve consistency. However, it may not be optimal for more challenging multi-turn tasks. Importantly, this comparison highlights the effectiveness of multimodal in-context learning in advancing image generation and editing.
>
> ---
> > (W4) the segmentation prediction task helps stabilize the movement of the subjects
>
> Thanks for the question. Yes. The current segmentation mask is always “the same mask as in the previous editing round”, and predicting it helps the model align the camera views and subject positions, stabilizing the movement and enhancing consistency.
> Besides, we can also append the predicted segmentation masks as conditions to increase more constraint tokens to achieve better consistency, as discussed in W2 and W3.
>
> ---
> > (W5) full attention and block-wise causal attention
>
> Thank you for pointing out this potential source of confusion. In this work, we *independently* implemented the two attention variants and *trained separate models* in the early stage, as illustrated in Fig. 11, with corresponding results reported in Table 10.
>
> We observe that full attention outperforms block-wise causal attention, likely due to its bidirectional perceptual field and higher computation complexity. Furthermore, our initial video foundation model was pre-trained using full attention, which informed our decision to adopt this variant for continued training on interleaved data—particularly given our limited computational resources. Accordingly, *the final version of VINCIE is implemented with **full attention***.
>
> Despite this, we believe that block-wise causal attention remains a promising direction. Its naturally causal temporal structure aligns well with real-world dynamics and offers practical benefits such as compatibility with key-value (KV) caching for efficient inference. We have clarified the description of the attention mechanism in the revised manuscript to improve clarity.

---

> ### Author Response · Authors · 2025-11-21
> **Author Response to Reviewer 6Gue (3/3)**
>
> > (W6.1) More detailed descriptions for input conditions and tasks.
>
> Thanks a lot for the considerate suggestion. We have enumerated all possible conditions of our model in **Fig. 12 in Appendix**.
>
> > (W6.2) Right now it is a bit confusing on how the model operates during inference: for example, when adding the segmentation mask prediction task during inference, will the predicted segmentation mask be used as part of the input condition to the model, or will it only be predicted, but not used during the next edited image prediction?
>
> During inference, the model operates in an autoregressive manner: it first predicts the segmentation mask, which is then appended as part of the input condition for the subsequent generation, i.e., source image → segmentation mask → target image, at each turn.
>
> ---
> > Overall, I find the method and results to be promising, but I believe the current paper's presentation can be improved. I would be happy to raise my score if the concerns can be addressed.
>
> Finally, we would like to sincerely thank you for the thorough review and constructive comments. We have revised the paper accordingly and carefully proofread it to improve clarity and presentation.

---

> ### Author Response · Authors · 2025-11-25
> **Follow-Up on Discussion**
>
> Dear Reviewer 6Gue,
>
> Thank you once again for your thoughtful feedback and the time devoted to reviewing our submission. We have carefully addressed the concerns raised and provided detailed responses in this rebuttal.
>
> If any points remain unclear or require further clarification, we would be very happy to provide additional explanation or engage in further discussion.

---

> ### Author Response · Authors · 2025-11-26
> **Summary of Author Response**
>
> Thank you again for your detailed and constructive feedback. We have carefully revised the manuscript and provided comprehensive responses to all your comments. In particular, we have:
>
> - **W1**: Clarified the training objectives for CSP and NSP, and unified them under the denoising MSE formulation.
>
> - **W2**: Provided intuition and task-level analysis explaining why CS→NS→I benefits MagicBrush but slightly underperforms on MSE-Bench; added per-category PF/SC/OV breakdowns to support the explanation.
>
> - **W3**: Extended the comparison between Dummy Context and History Context on MSE-Bench across five turns, showing that Dummy Context helps early turns but degrades in later, more complex edits.
>
> - **W4**: Elaborated on why segmentation prediction stabilizes camera and subject movement, clarifying the role of predicted CS/NS as structural constraints.
>
> - **W5**: Clarified that the final version of VINCIE uses full attention, explained why, and improved the presentation to avoid confusion with block-wise causal attention.
>
> - **W6**: Expanded the inference description, enumerated all input-condition modes in the Appendix, and clarified how predicted segmentation masks are fed into subsequent generation steps.
>
> We hope these revisions have fully addressed your concerns. If any points remain unclear or require further clarification, we would be very happy to provide additional details.
>
> Thank you again for your time and thoughtful feedback.

---

> ### Comment · Reviewer_6Gue · 2025-11-27
>
> Dear authors,
>
> Thanks for the rebuttal. I think the additional experiments on NS vs. CS+NS prediction during inference and the Dummy context vs. Editing history comparison make sense.
>
> I have one additional question regarding the training objectives of NSP and CSP. It is mentioned in the rebuttal that NSP and CSP are formulated as the denoising objective over the segmentation masks. What is the format of the segmentation mask? Is it in the format of VAE latents encoded from the RGB images of the segmentation mask, with the three RGB channels being identical? Or is it a latent feature encoded from gray-scale segmentation mask images?
>
> I would strongly recommend adding these detailed descriptions of the NSP and CSP formulations into the main paper, instead of only using descriptions like "NSP incorporates the current segmentation map into the context" (line 265).
>
> Finally, can the author provide some intuition on why this denoising objective brings benefits, compared to e.g. simply adding an MLP layer after the output visual features to directly predict the segmentation mask in pixel space? From my understanding, in order to use the predicted segmentation mask autoregressively during inference, the segmentation mask also needs to be decoded from latent into pixel space images, which increases the inference complexity. Any intuition could be helpful and no additional experiments are required here.
>
> Since the final version of VINCIE uses full attention, please also update line 94 in the manuscript ("Next, we design block-wise causal attention within a diffusion transformer...") to better reflect the actual model design.

---

> > ### Author Response · Authors · 2025-11-27
> >
> > Dear Reviewer 6Gue,
> >
> > Thanks for your response and for the positive feedback regarding our experiments for NS vs. CS+NS and Dummy context vs. Editing history. We address your concerns as follows.
> >
> > We choose the former one, i.e., `in the format of VAE latents encoded from the RGB images of the segmentation mask, with the three RGB channels being identical`.
> >
> > Thanks very much for your recommendation. We have added this implementation detail to Sec. 3.3 of the main paper, highlighted in magenta.
> >
> > Regarding the benefits of the denoising objective, we provide the following justifications:
> >
> > - Compared to simply adding an MLP layer to predict segmentation masks, unifying segmentation and image generation with the generative modeling via the denoising objective facilitates **ability transfer** from segmentation prediction to image generation—particularly for grounding and layout planning, as discussed above.
> > - This unification strategy also provides promising **scalability to other modalities**, such as depth maps, surface normals, canny edges, and keypoints/landmarks, which paves the way for future work. Unifying these modalities in a generative manner instead of introducing modality-specific heads (e.g., MLP layers) offers a more elegant and generalizable design.
> > - In fact, it is computationally efficient, as we can directly append the generated segmentation latent into context in subsequent generation steps, **no need for decoding them back into pixel space images**-unless user-side intervention is required, which is uncommon in practice.
> >
> > We have also updated the manuscript around line 95 (highlighted in magenta) to more accurately reflect the actual model design.
> >
> > Thank you again for your constructive suggestions, which have meaningfully improved the clarity and depth of our manuscript.

---

> > > ### Comment · Reviewer_6Gue · 2025-11-28
> > >
> > > Thanks for the responses, my concerns have been addressed. I would recommend including the Dummy context vs. Editing history comparison in the revision (could be helpful to include them in the appendix if there is not enough space in the main text).
> > >
> > > I'm happy to raise my score but it seems that currently the system has disabled editing the reviews. I will try to see if I can update it later.

---

> ### Author Response · Authors · 2025-11-28
>
> **Thank you for your thoughtful review and for noting that you will raise the score**. We appreciate your positive evaluation and the time you dedicated to our submission.
>
> We will organize the recommended results, including the Dummy context vs. Editing history comparison, and include these results in the final version.

---

### Official Review · Reviewer_yqUb · 2025-10-30

**Soundness:** 3
**Presentation:** 3
**Contribution:** 3
**Rating:** 8
**Confidence:** 5

**Summary:**

VINCIE introduces a novel approach to multi-turn image editing by directly leveraging video data. It treats sequential video frames and their natural transitions as in-context editing demonstrations, and designed three proxy tasks: next-image predic-
047 tion, current segmentation prediction, and next-segmentation prediction to tickle this problem. Using a diffusion transformer architecture guided by predicted segmentation masks, VINCIE learns to localize and apply edits step-by-step in a context-aware manner without needing manually curated editing datasets. The method shows strong performance on new and established multi-turn editing benchmarks.

**Strengths:**

S1) This paper proposed a new perspective in constructing session-wise data with long, interleaved image-text context from native videos while prior works that used video for editing were mainly for constructing pair-wise data. This highlights the originality.

S2) At first glance the designed approach is best suited for videos with a static viewpoint. When the camera moves, the region proposal and segmentation process can break down, leading to inaccurate edits or mask predictions. Large background or subject position shifts caused by camera movement are not explicitly addressed. But it is clever approach that predicting segmentation masks before editing could mitigates subject drift.

S3) The paper is well written, with good motivation, and communicated clearly. Overall I enjoyed reading this work.

**Weaknesses:**

W1) I find it surprising that the paper identifies MagicBrush (NeurIPS 2023) as the last major benchmark for multi-turn image editing, with no subsequent progress. Based on my recollection, ImgEdit-Bench (May 2025) also introduced evaluation protocols on multi-turn image editing. Including results on the ImgEdit-Bench benchmark would make the empirical comparison much more convincing.

**Questions:**

Q1) In Visual Transition Annotation, authors leveraged VLM for captioning. While it makes sense to me to use VLM for generating detailed and coherent descriptions of each frame from multiple aspects, I am concerned of using VLMs to identify semantic and visual differences
between the two frames when the two frames have minor differences. It is known that VLMs often fails in telling subtle difference between two images, as pointed out in The Dawn of LMMs and VIEScore. I question the robustness of this approach for fine-grained visual transition annotation, as critical details might be overlooked or misrepresented when using these models. Could the authors provide a case study or qualitative results focusing on more fine-grained edits (e.g. moving an object slightly to one direction, like only 1cm)?

Q2) Regarding Regions-of-Editing, GroundingDINO and SAM2 may propagate errors in mask generation or transition description, especially in complex or cluttered scenes. Any measures are conducted to minimize the error impact in this pipeline?

---

> ### Author Response · Authors · 2025-11-21
> **Author Response to Reviewer yqUb**
>
> We sincerely thank the reviewer for the highly insightful evaluation and constructive critique. We consider the feedback to be invaluable to the improvement of this work, and we have addressed each point in detail below.
>
> ---
>
> > (W1) ... Including results on the ImgEdit-Bench benchmark would make the empirical comparison much more convincing.
>
> Thanks a lot for your constructive comments. The reason we did not adopt ImgEdit-Bench in our submission is that only source images and editing instructions in ImgEdit-Bench were released, but **the detailed scoring method is missing**.
>
> To achieve the evaluation on this benchmark, we resort to our own scoring strategy in MSE-Bench. In detail,
> - we assess PF (Prompt following) and SC (semantic consistency), and Overall (the geometric mean of PF and SC);
> - we use a more fine-grained scale of 0 to 10;
> - instead of evaluating all turns, we call API once for each turn, so that each call focus on one turn with previous turns as context. All of the instruction templates have been added in Sec. C. 2 in Appendix.
>
> The results shown in the table below demonstrate the effectiveness and superiority of the proposed method, consistent with the performance comparison in MSE-Bench.
>
> | Method | Turn1 | | | Turn2 | | | Turn3 | | |
> |--------|------|-----|------|------|-----|------|------|-----|------|
> |        | PF   | SC  | OV   | PF   | SC  | OV   | PF   | SC  | OV   |
> | Nano Banana | 9.49 | 9.42 | 9.41 | 9.07 | 8.62 | 8.80 | 8.37 | 7.51 | 7.78 |
> | - | - | - | - | - | - | - | - | - | - |
> | ICEdit | 7.93 | 8.80 | 8.03 | 5.91 | 6.20 | 5.96 | 3.10 | 4.45 | 3.43 |
> | Omnigen | 7.27 | 8.86 | 7.78 | 5.19 | 5.60 | 5.28 | 2.67 | 3.55 | 2.80 |
> | Omnigen2 | 7.74 | 9.04 | 7.89 | 3.87 | 5.73 | 4.23 | 1.49 | 2.15 | 1.67 |
> | Bagel | 8.52 | 8.83 | 8.60 | 6.52 | 7.02 | 6.59 | 4.23 | **5.54** | 4.45 |
> | FLUX.1-Kontext | 8.09 | **9.26** | 8.15 | 6.60 | 6.77 | 6.56 | 3.82 | 5.43 | 4.23 |
> | Qwen-Image-Edit | **8.82** | 9.11 | **8.88** | 6.88 | 7.13 | 6.80 | 3.61 | 4.63 | 3.66 |
> | Ours | 8.44 | 9.08 | 8.58 | **6.93** | **7.19** | **6.97** | **4.69** | 5.30 | **4.74** |
>
>
> ---
> > (Q1) Concerns for VLMs to identify subtle semantic and visual differences
>
> Thank you for the insightful questions. We agree that current VLMs still face challenges in capturing subtle visual differences. However, this limitation can be partially mitigated by incorporating grounding signals—for instance, by generating descriptions based not only on visual frames but also on differences in segmentation maps, bounding boxes, or depth maps between frames.
>
> Our work primarily focuses on modeling significant semantic changes and enabling in-context generation and editing, while acknowledging the handling of subtle variations as an important direction for future work.
>
> To further address this concern, we provide a case study based on the recommended editing example (see **Fig. 25 in Appendix**). The results show that even the strongest model, Nano Banana, struggles with subtle editing cues. In contrast, our method supports flexible the input condition (i.e., darg) to achieve fine-grained editing. We believe that deeper investigation into this challenge could significantly advance the capabilities of image editing and visual content creation.
>
> ---
> > (Q2) potential error propagation risk brought by GroundingDINO and SAM2
>
> To mitigate potential error propagation, we have adopted the following filtering strategies:
>
> - **Detection confidence filtering**: We extracted object-wise confidence scores from GroundingDINO and discarded detections below a predefined threshold.
>
> - **Description-segmentation alignment**: We computed the CLIP text–image (T-I) score between each object description and its cropped region, filtering out samples with low semantic alignment.
>
> - **Cross-frame consistency**: We measured CLIP image–image (I-I) scores between cropped objects across frames, retaining only those with high visual consistency.

---

> > ### Comment · Reviewer_yqUb · 2025-11-24
> >
> > Thanks for the rebuttal. For W1, it would be more reasonable to justify in the main paper when introducing magicbrush. measures used in Q1/2 also sound reasonable to me. I will keep the positive rating.

---

> > > ### Author Response · Authors · 2025-11-25
> > >
> > > Thank you for your thoughtful feedback! Following your suggestion, we have added a justification in the main paper (Section 4.3, Paragraph 2) explaining our choice of MagicBrush:
> > > > "Given its support for multi-turn editing, high-quality manual annotations, and close alignment with real-world editing needs, we first adopt MagicBrush to evaluate our method and compare against baselines."
> > >
> > > In addition, we will include the results on ImgEdit-Bench, along with discussions addressing Q1 and Q2, in the revised manuscript.
> > >
> > > We sincerely appreciate your valuable comments, which have significantly contributed to improving the quality and clarity of our work.

---

### Official Review · Reviewer_bA7V · 2025-10-31

**Soundness:** 3
**Presentation:** 3
**Contribution:** 2
**Rating:** 4
**Confidence:** 4

**Summary:**

This paper proposes VINCIE, a method for learning in-context image editing directly from video data. The key innovation is constructing training data by: (1) sampling frames from videos, (2) using VLMs to annotate visual transitions between frames, (3) generating segmentation masks for regions of interest using GroundingDINO and SAM2, and (4) training a diffusion transformer with three proxy tasks (next-image prediction, current segmentation prediction, next segmentation prediction). The authors introduce MSE-Bench, a challenging 5-turn editing benchmark, and demonstrate competitive performance against existing methods while showing scalability with increasing data.

**Strengths:**

1. Well-structured with clear figures (Figures 1-3 effectively convey main ideas)
2. Comprehensive appendix with implementation details
3. Three proxy tasks (NIP, CSP, NSP) provide complementary learning signals
4. Comprehensive ablation studies validate design decisions

**Weaknesses:**

1. The claim of learning from "native videos" is misleading—the method requires extensive preprocessing with VLMs, GroundingDINO, and SAM2. This is annotation-based learning, not purely native video learning
2. Evaluation relies heavily on GPT-4o as judge, which may introduce bias despite correlation analysis (Table 7)
3. Core technical novelty is limited: DiT architecture, segmentation prediction, and in-context learning are established techniques. The main contribution is the data construction pipeline
4. Heavy reliance on existing models (VLM, GroundingDINO, SAM2) reduces the "learning from video" novelty
5. Fair comparison at same dataset?

**Questions:**

see the Weaknesses.  Overall, I would not object to this paper being accepted.

---

> ### Author Response · Authors · 2025-11-21
> **Author Response to Reviewer bA7V (1/2)**
>
> We sincerely thank the reviewer for the thoughtful evaluation and constructive feedback. Your comments are highly valuable to us. Below, we provide a point-by-point response addressing each concern.
>
> ---
>
> > (W1) The claim of learning from "native videos" is misleading—the method requires extensive preprocessing with VLMs, GroundingDINO, and SAM2. This is annotation-based learning, not purely native video learning
>
> Thanks for raising this concern. In our paper, “native videos” refers to the **natural video data as the visual modality**, comparable to existing image editing datasets [f] that heavily depend on synthesized images with low quality or artifacts. We will polish this description and make it clearer.
>
> As for the annotation-base learning concern, we would like to clarify that the aim of employing these preprocessing steps is to **bridge the semantic and structural gap between two sampled frames, instead of providing targets or supervision signals in annotation-based learning**.  This direction has been verified effective and promising by recent well-known works, like DALL-E 3 [g] and LLaVA [h], and Qwen-VL-Chat[i].
>
> [f] OmniEdit: Building Image Editing Generalist Models Through Specialist Supervision. ICLR'25
>
> [g] Improving Image Generation with Better Captions. 2024
>
> [h] Visual Instruction Tuning. NeurIPS'23
>
> [i] Qwen-VL: A Versatile Vision-Language Model for Understanding, Localization, Text Reading, and Beyond. arXiv'23
>
> ---
> > Evaluation relies heavily on GPT-4o as judge, which may introduce bias despite correlation analysis (Table 7)
>
> We acknowledge the limitations of current GPT-based evaluation methods but note that they remain the most viable option at present. Existing benchmarks—such as VIEScore [d], GEdit-Bench, and ImgEdit [e]—also rely on GPT-based evaluation, reflecting its widespread adoption. Furthermore, the correlation analysis in Table 7 shows that GPT-based metrics align more closely with human judgment compared to other automatic metrics, indicating relatively higher reliability.
>
> To mitigate potential model bias, we additionally evaluated our method and all baselines using **Gemini-2.5-Pro** and **Doubao-Seed-1.6-Vision**. While different evaluators may exhibit varying levels of strictness, the results remain consistent with those obtained using GPT-4o, further validating the robustness and effectiveness of our approach.
>
> | Method | Gemini-2.5-Pro |  |  | Doubao-Seed-1.6-Vision |  |  |
> |--------|--------|------|------|--------|------|------|
> |        | Turn3 | Turn4 | Turn5 | Turn3 | Turn4 | Turn5 |
> | GPT-4o | 66.67 | 55.67 | 47.00 | 84.33 | 82.67 | 77.67 |
> | Nano Banana | 69.00 | 61.67 | 49.00 | 83.33 | 83.00 | 79.67 |
> | - | - | - | - | - | - | - |
> | ICEdit | 4.67 | 0.33 | 0.00 | 20.33 | 13.00 | 8.33 |
> | Omnigen2 | 3.33 | 0.67 | 0.33 | 33.67 | 25.00 | 17.00 |
> | Step1X-Edit | 15.67 | 2.67 | 1.33 | 41.00 | 28.33 | 18.67 |
> | Bagel | 28.33 | 11.33 | 2.00 | 64.00 | 52.00 | 40.00 |
> | FLUX.1-Kontext | 29.67 | 17.33 | 10.00 | 59.33 | 53.00 | 45.00 |
> | Qwen-Image-Edit | **40.67** | 16.00 | 7.33 | 66.33 | 57.67 | 43.67 |
> | Ours | 34.67 | **22.67** | **10.33** | **67.67** | **62.67** | **54.33** |
>
> ---
> > (W3) Core technical novelty is limited: DiT architecture, segmentation prediction, and in-context learning are established techniques. The main contribution is the data construction pipeline
>
> In this work, we do not aim to explore new model architecture, but to propose a new **architecture-agnostic training paradigm** to model the interleaved session data from videos, to unlock in-context image generation and editing abilities.
>
> Different from existing work [j] which simply introduces segmentation prediction as an auxiliary learning objective, we are the first to explicitly investigate the distinct roles of **current and next segmentation modeling**, which serve as condition and target for controllable generation and grounding, respectively. Furthermore, we also explore the possibility of **treating segmentation maps intermediate states for “Chain-of-Editing”**, decomposing complex generation or editing tasks into a sequence of simpler transformations (source image → source seg. → target seg. → target image), as shown in **Fig. 12 (Appendix)**.
>
> As for in-context learning, existing work mainly focuses on CoT with pure text [k] or transfering visual relations [l] for single-turn editing, In contrast, our method uniquely leverages **multimodal interleaved context** to tackle more complex and multi-turn generative tasks.
>
>
> [j] Emu Edit: Precise Image Editing via Recognition and Generation Tasks. CVPR'24
>
> [k] ImageGen-CoT: Enhancing Text-to-Image In-context Learning with Chain-of-Thought Reasoning. ICCV'25
>
> [l] Edit Transfer: Learning Image Editing via Vision In-Context Relations. arXiv'25

---

> ### Author Response · Authors · 2025-11-21
> **Author Response to Reviewer bA7V (2/2)**
>
> > (W4) Heavy reliance on existing models (VLM, GroundingDINO, SAM2) reduces the "learning from video" novelty
>
> We would like to clarify the novelty of this work as follows:
> - We propose **a scalable learning framework** that pretrains on large-scale **interleaved multimodal data**, with **next-frame, current-segmentation, and next-segmentation prediction** as learning objectives.
> - The interleaved data, the learning framework, and the three objectives **unlock more advanced abilities** beyond prior single-turn editing, such as multi-turn editing, multi-concept composition, and story generation.
> - We explore **segmentation modeling** as conditions and targets, **unlocking controllable editing and chain-of-editing**. To the best of our knowledge, we are the first to make all of the above explorations.
>
> As described in W1, these existing models (VLM, GroundingDINO, SAM2) just help bridge the semantic and structural gap between two sampled frames. Other well-known works, like DALL-E 3 [g] and LLaVA [h], also rely on prior models to automatically achieve this goal.
>
> Besides, the proposed method does not heavily rely on these existing models; instead, it is **orthogonal to these models**, i.e., advancements of understanding models in the future will further support the effectiveness and superiority of the proposed method.
>
> ---
> > (W5) Fair comparison at same dataset?
>
> To strictly ensure the fairness of our evaluation and comparison with baseline models, we reproduce all of the baselines and keep the inference hyperparameters (such as random seeds, denoising steps) all the same. More implementation details have also been reported in Sec. 4.1 and in our released code.

---

> ### Author Response · Authors · 2025-11-25
> **Follow-Up on Discussion**
>
> Dear Reviewer bA7V,
>
> Thank you once again for your thoughtful feedback and the time devoted to reviewing our submission. We have carefully addressed the concerns raised and provided detailed responses in this rebuttal.
>
> If any points remain unclear or require further clarification, we would be very happy to provide additional explanation or engage in further discussion.

---

> ### Author Response · Authors · 2025-11-26
> **Summary of Author Response**
>
> Thank you again for your detailed and constructive feedback. We have carefully reviewed your comments and provided comprehensive responses. In particular, we have:
>
> - **W1**: Clarified the meaning of “native videos,” refined the terminology, and explained that VLM/GroundingDINO/SAM2 are used only to bridge semantic–structural gaps rather than to provide annotation-style supervision.
>
> - **W2**: Addressed concerns about GPT-based evaluation by adding results from **Gemini-2.5-Pro** and **Doubao-Seed-1.6-Vision**, confirming consistent trends across evaluators.
>
> - **W3**: Highlighted the core contribution as a new **architecture-agnostic interleaved-video training paradigm**, and clarified the distinct roles of current vs. next segmentation modeling, including chain-of-editing capabilities.
>
> - **W4**: Clarified that the method does not rely on specific pretrained models; rather, these tools are orthogonal and only assist in constructing interleaved context.
>
> - **W5**: Ensured fairness by reproducing all baselines under identical inference configurations and reporting details in Sec. 4.1 and the released code.
>
> We hope these clarifications have fully addressed your concerns. If any points remain unclear or you would like further detail, we would be happy to elaborate.
>
> Thank you again for your time and thoughtful feedback.

---

> ### Author Response · Authors · 2025-11-27
>
> Dear Reviewer bA7V,
>
> Thank you for your thoughtful comments and careful review. We have addressed each point in our rebuttal.
>
> If any part is still unclear, we’re glad to elaborate. We would be grateful if you could re-evaluate the paper considering these revisions.

---

> > ### Comment · Reviewer_bA7V · 2025-11-28
> >
> > Thank you for your detailed rebuttal. These clarifications have addressed most of my previous concerns. Therefore, I am raising my recommendation score to 6.

---

> ### Author Response · Authors · 2025-11-28
>
> **Thank you for your insightful review and for informing us that you intend to raise the score.** We greatly appreciate your thorough evaluation and constructive comments.

---

### Official Review · Reviewer_wj9T · 2025-11-01

**Soundness:** 3
**Presentation:** 3
**Contribution:** 3
**Rating:** 4
**Confidence:** 4

**Summary:**

This paper introduces VINCIE, a framework for in-context image editing using video data. The core premise is to learn this capability solely from native video data, bypassing the need for curated, paired image-editing datasets. The authors propose a scalable data pipeline that samples video frames and uses a Vision-Language Model to generate textual annotations of the visual transitions between them. It also generates segmentation masks for these "Regions of Editing" (RoEs). The model, a diffusion transformer, is trained using three proxy tasks: Next-Image Prediction (NIP), Current Segmentation Prediction (CSP), and Next-Segmentation Prediction (NSP). To evaluate their method, the authors introduce a new multi-turn benchmark, MSE-Bench, which consists of 5-turn editing sessions and is evaluated using GPT-4o. The results demonstrate that VINCIE achieves strong performance, outperforming many academic methods on MagicBrush and their new benchmark, and that the approach scales well with data.

**Strengths:**

1. The central idea of using native video data to learn in-context editing makes sense. It cleverly reframes the problem, identifying videos as a natural, abundant, and scalable source of "edit" data (i.e., state transitions) that intrinsically contains the visual consistency and temporal context missing from static image-pair datasets.

2. The data construction pipeline is sound, combining a VLM for high-level semantic transition annotation with grounding models (GroundingDINO + SAM2) for precise, low-level spatial localization (RoEs). The three proxy tasks (NIP, CSP, NSP) are well-motivated; using segmentation prediction not only as a training signal (CSP) but also as a predictive target (NSP) is a smart way to improve grounding and controllable generation.

3. The paper is well-written, and the methodology is presented clearly. The figures are effective at illustrating the data pipeline and model architecture.

**Weaknesses:**

1. The paper defines in-context editing as modifying an image based on a "contextual sequence comprising text and previously generated images." This framing is functionally equivalent to multi-turn editing, where the primary role of the context is to serve as the input for the next step. This definition is quite narrow and overlooks a more common interpretation of "in-context" for image models: the ability to use one or more reference images in the context to provide new subjects, styles, or concepts for the current edit (e.g., "add the dog from the second image with the style from the 4th image into this one"). The paper's experiments all fit the multi-turn definition, and it's unclear if VINCIE can perform this more complex, composition-focused "reference-based" in-context editing.

2. The evaluation, while thorough on MagicBrush and the proposed MSE-Bench, feels incomplete. It would be much more convincing to see a comparison on more recent benchmarks, such as GEdit-Bench and also evaluate on the EMUEdit benchmark?

3. The primary evaluation for the newly proposed MSE-Bench relies on a "success rate" computed by GPT-4o. This is a significant concern. The reliability of current VLMs for nuanced, multi-step comparative judgment is questionable. The appendix suggests multiple images are fed simultaneously, which could easily confuse the model. My own experience suggests GPT-4o can struggle to correctly identify subtle edits or preserve consistency even in simple source-vs-target comparisons, let alone a complex 5-turn sequence where errors propagate. This evaluation method needs much stronger validation.
Also, for automatic evaluation, it would be beneficial to also report scores like the PF (Prompt following) and SC (semantic consistency) from recent work like VIEScore: Towards Explainable Metrics for Conditional Image Synthesis Evaluation. These are designed for image editing and might offer a more reliable supplement to the GPT-4o-based success rate.

**Questions:**

Natural videos excel at capturing state transitions like object addition/removal, pose changes, and motion. However, they rarely, contain examples of more "creative" or "unrealistic" edits that are common user requests. For example, drastic environmental changes (e.g., "change the weather from snow to summer"), complex style transfers (e.g., "make this look like a Picasso painting"), or material transformations (e.g., "make the person out of glass"). It is questionable whether a model trained only on natural video can generalize to these common editing tasks. I do agree that video data could be used in the pretraining, but using video data for image editing is not a novel concept. The paper's own related work section cites several recent methods that also "Learn from Video for Image Generation." The paper differentiates itself by using "long, interleaved image-text context" rather than just frame pairs, but this feels more like a strong incremental improvement than a paradigm shift.

---

> ### Author Response · Authors · 2025-11-21
> **Author Response to Reviewer wj9T (1/2)**
>
> We sincerely thank the reviewer for the thorough evaluation and helpful comments. Your feedback is highly valuable to us, and we respond to each weakness and question point-by-point in the following.
>
> ---
>
> > (W1.1) The definition of in-context image editing.
>
> Thanks for raising this concern. As shown in **Fig. 1, Fig. 19 - Fig. 25 (Appendix)**, our model is able to perform other in-context image editing or generation (e.g., controllable editing, chain-of-editing, reference-based image composition, and story keyframe generation), rather than only Multi-Turn Image Editing (MTIE). We focus on MTIE in this work due to the missing or inadequate evaluation methods for other tasks. We will carefully revise the definition of in-context image editing to make it clearer in the updated submission.
>
> > (W1.2) composition-focused "reference-based" in-context generation and editing
>
> Thanks for your constructive comments on **composition-focused "reference-based" in-context editing**. We add some cases in **Fig. 20 in Appendix**, which demonstrates that our model already has this ability.  Besides, we also added a performance comparison with recent work on  the **in-context image generation** task (MULTIPLE task type) from the OmniContext benchmark, as shown in the table below (Char: Character. Obj: Object. C + O: Character and Object. Avg: Average. PF: Prompt Following. SC: Subject Consistency. OV: Overall. Avg: Average).
> The results demonstrate that our method not only possesses strong in-context generation capabilities but also outperforms recent baselines.
>
>
> | Method | Char (PF) | Char (SC) | Char (OV) | Obj (PF) | Obj (SC) | Obj (OV) | C+O (PF) | C+O (SC) | C+O (OV) | Avg (PF) | Avg (SC) | Avg (OV) |
> | :--- | :---: | :---: | :---: | :---: | :---: | :---: | :---: | :---: | :---: | :---: | :---: | :---: |
> | GPT-4o | 9.17 | 9.03 | 9.07 | 9.06 | 8.90 | 8.95 | 8.34 | 8.89 | 8.54 | 8.86 | 8.94 | 8.86 |
> | - | - | - | - | - | - | - | - | - | - | - | - | - |
> | UNO [a] | 3.88 | 2.38 | 2.54 | 7.46 | 5.86 | 6.51 | 5.10 | 4.10 | 4.39 | 5.48 | 4.11 | 4.48 |
> | OmniGen [b] | 5.92 | **6.18** | 5.65 | 5.60 | 5.46 | 5.44 | 4.64 | 4.96 | 4.68 | 5.39 | 5.53 | 5.26 |
> | BAGEL [c] | 6.14 | 4.86 | 5.17 | **7.54** | 6.10 | 6.64 | **6.74** | 6.28 | **6.24** | **6.81** | 5.75 | 6.02 |
> | Ours | **6.44** | 6.12 | **5.72** | 7.02 | **7.02** | **6.66** | 5.80 | **6.60** | 5.79 | 6.42 | **6.58** | **6.05** |
>
>
>
> [a] Less-to-More Generalization: Unlocking More Controllability by In-Context Generation. ICCV'25
>
> [b] OmniGen: Unified Image Generation. arXiv'24
>
> [c] Emerging Properties in Unified Multimodal Pretraining. arXiv'25
>
>
> ---
> > (W2) Performance comparison on GEdit-Bench and EMUEdit.
>
> Thank you for the suggestion. In our original submission, we did not include the two benchmarks as our work focuses on in-context image editing, particularly in the multi-turn setting, whereas both benchmarks are limited to single-turn editing scenarios.
>
> We also agree that including comparisons on these two benchmarks would further strengthen our work. Here, we evaluated and compared baselines and our method on them. Due to the limited time, we only evaluate on the representative categories.
> The results on **GEdit-Bench** are reported in the following tabel. SQ (Semantic Consistency), PQ (Perceptual Quality), OV (Overall Score).
>
> | Method | remove |  |  | add |  |  | background |  |  | color |  |  |
> |--------|--------|----|----|------|----|----|--------|----|----|--------|----|----|
> |        | SQ | PQ | OV | SQ | PQ | OV | SQ | PQ | OV | SQ | PQ | OV |
> | ICEdit | 4.86 | 6.83 | 4.76 | 6.02 | 6.73 | 5.88 | 4.83 | 6.85 | 4.98 | 7.50 | 6.20 | 6.40 |
> | OmniGen2 | 6.42 | 6.54 | 6.17 | 6.62 | 7.07 | 6.33 | 7.48 | 6.73 | 6.73 | 6.88 | **6.38** | 6.38 |
> | Flux_Kontext | 6.75 | 6.90 | 6.21 | **7.38** | 7.05 | 6.78 | 7.18 | **7.13** | 6.83 | **8.10** | 6.35 | **7.06** |
> | Bagel | 6.63 | 6.39 | 6.27 | 7.12 | 7.25 | 6.78 | **7.60** | 6.60 | 6.88 | 7.48 | 6.18 | 6.68 |
> | Ours | **7.11** | **7.14** | **6.52** | 7.32 | **7.32** | **6.95** | 7.28 | 6.90 | **6.93** | 7.28 | 6.33 | 6.63 |
>
> As for **EMUEdit**, we adopted the scoring method proposed in VIEScore [d] for more trustworthiness. The results are reported as follows.
>
> | Method | add | background | color | local | remove |
> |--------|------|-------------|--------|--------|---------|
> | ICEdit   | 5.82 | 3.86        | 5.70   | 5.02   | 4.42    |
> | OmniGen2   | 4.25 | 4.21        | 5.21   | 5.24   | 6.12    |
> | Flux_Kontext   | 6.21 | 4.87        | 5.93   | 5.96   | **6.30**    |
> | Bagel   | 6.08 | 4.54        | **6.07**   | **6.15**   | 6.24    |
> | Ours   | **6.56** | **5.28**   | 5.81   | 5.36   | 6.23    |
>
> Results show that our method still performs well compared with others, showing **its remarkable basic editing ability in the single-turn editing scenario**.
>
> [d] VIEScore: Towards Explainable Metrics for Conditional Image Synthesis Evaluation. ACL'24

---

> ### Author Response · Authors · 2025-11-21
> **Author Response to Reviewer wj9T (2/2)**
>
> > (W3.1) The reliability of current VLMs for nuanced, multi-step comparative judgment
>
> Thanks for raising this concern. We agree that current GPT-based evaluation methods are imperfect, but we may have no better choice but to make this comprise. On the one hand, current commonly used benchmarks, such as VIEScore [d], GEdit-Bench, and ImgEdit [e], also rely on GPT-based evaluation. On the other hand, the correlation between automatic metrics and human evaluation in Tab. 7 demonstrates that GPT-based evaluation is much more reliable than other automatic metrics.
>
> [e] ImgEdit: A Unified Image Editing Dataset and Benchmark. NeurIPS’25
>
> > (W3.2) Concerns for evaluation on 5 turns on one pass and reporting scores like the PF (Prompt following) and SC (semantic consistency)
>
> Thanks for your very constructive advice! According to it, we made further improvements on our GPT-based evaluation:
> - assess **PF (Prompt following) and SC (Semantic Consistency), and Overall (the geometric mean of PF and SC)**;
> - a more **fine-grained scale of 0 to 10**;
> - instead of evaluating all turns in one pass, we call API once for each turn, so that **each API call focuses on one turn** with previous turns as context.
>
> All of the instruction templates have been added in Sec. C.6 of Appendix.  The results are shown below, with a similar observation that **our method performs better than recent baselines, especially on more turns of editing**.
>
>
> | Method | Turn3 |  |  | Turn4 |  |  | Turn5 |  |  |
> |--------|-------|----|----|-------|----|----|-------|----|----|
> |        | PF    | SC | OV | PF    | SC | OV | PF    | SC | OV |
> | GPT-4o   | 8.46  | 8.09 | 8.21 | 8.25 | 7.69 | 7.93 | 7.71 | 7.48 | 7.51 |
> | Nano Banana   | 8.75 | 8.33 | 8.49 | 8.50 | 7.95 | 8.20 | 7.89 | 7.80 | 7.78 |
> | - | - | - | - | - | - | - | - | - | - |
> | ICEdit | 2.22 | 2.71 | 2.30 | 1.08 | 1.62 | 1.19 | 0.56 | 0.81 | 0.62 |
> | Omnigen2 | 2.97 | 3.52 | 3.04 | 1.81 | 2.15 | 1.84 | 1.06 | 1.39 | 1.15 |
> | Step1X-Edit | 3.97 | 4.58 | 3.97 | 2.44 | 2.85 | 2.50 | 1.17 | 1.66 | 1.28 |
> | Bagel | 6.52 | 6.32 | 6.33 | 5.39 | 5.34 | 5.23 | 4.35 | 4.30 | 4.25 |
> | FLUX.1-Kontext | 6.27 | 6.29 | 6.17 | 5.18 | 5.36 | 5.12 | 4.26 | 4.52 | 4.30 |
> | Qwen-Image-Edit | **7.14** | **7.48** | **7.06** | **5.85** | **5.95** | **5.74** | 4.43 | 4.66 | 4.35 |
> | Ours | 6.82 | 6.97 | 6.73 | 5.67 | **5.95** | 5.63 | **4.50** | **4.80** | **4.55** |
>
>
> ---
> > (Q1.1) whether a model trained only on natural video can generalize to common editing tasks, such as drastic environmental changes, complex style transfers, and material transformations.
>
> Even though these scenarios are Uncommon in video data, we found that our model pre-trained on video data **could unlock these editing abilities and generalize to these tasks, as shown in Fig. 14 in Appendix**. The reasons may include:
> - Despite being uncommon, some scenarios (e.g. environmental changes) exist in our training data.
> - Our model is initialized with a video foundation model, and we also include text-to-video (T2V) as one of our pre-training tasks, which could help the model attain concepts like style and materials. These T2V abilities may be transferred to image editing.
>
>
> > (Q1.2) novelty and research contribution
>
> We would like to clarify that the focus on interleaved session data in this work goes beyond simply introducing more frames compared to pairwise video data. Rather, it aims to **enable more advanced capabilities in visual generative models, particularly in-context generation and editing**. This paradigm supports complex tasks such as *multi-reference composition, story keyframe generation, and chain-of-editing, and holds promise for future directions like multimodal chain-of-thought reasoning and world modeling*. These tasks can be unlocked by simple but effective context composition, as detailed in **Fig. 12 (Appendix)**.
> In contrast, pairwise data is inherently limited in scope, primarily supporting single-turn editing or basic customization.
>
> Similarly, this distinction parallels the trajectory of vision-language understanding. Despite the strong performance of recent VLMs (e.g., GPT-4o, GPT-5, Qwen3-VL), they continue to struggle with interleaved reasoning, often failing even basic multi-turn editing evaluations—a point we believe is mutually acknowledged in the above discussion (W3). Recent studies have also begun emphasizing the importance of interleaved understanding [e1, e2].
>
> [e1] CoMM: A Coherent Interleaved Image-Text Dataset for Multimodal Understanding and Generation. CVPR'25
>
> [e2] Emerging Properties in Unified Multimodal Pretraining. arXiv'25

---

> ### Author Response · Authors · 2025-11-25
> **Follow-Up on Discussion**
>
> Dear Reviewer wj9T,
>
> Thank you once again for your thoughtful feedback and the time devoted to reviewing our submission. We have carefully addressed the concerns raised and provided detailed responses in this rebuttal.
>
> If any points remain unclear or require further clarification, we would be very happy to provide additional explanation or engage in further discussion.

---

> ### Author Response · Authors · 2025-11-26
> **Summary of User Response**
>
> Thank you again for your detailed and constructive feedback. We have carefully revised the manuscript and provided comprehensive responses to all your comments. In particular, we have:
>
> - **W1.1 / W1.2**: Clarified the definition of *in-context image editing* and added new examples (Fig. 19–25) demonstrating broader capabilities such as reference-based composition, controllable editing, and story-driven multi-turn generation.
>
> - **W2**: Added new evaluations on **GEdit-Bench** and **EMUEdit**, showing that our method achieves strong performance in single-turn editing compared with recent baselines.
>
> - **W3.1 / W3.2**: Improved the evaluation protocol with per-turn GPT assessment, fine-grained PF/SC scoring, and updated prompt templates. We also added an ablation comparing evaluation strategies to support reliability.
>
> - **Q1.1**: Provided evidence that the model generalizes to creative and stylistic edits (e.g., weather, materials, style transfer), supported by new qualitative examples (Fig. 14).
>
> - **Q1.2**: Clarified the novelty of leveraging *interleaved image–text session data*, highlighting its role in enabling multi-reference composition, chain-of-editing, and more advanced in-context reasoning beyond pairwise video-training paradigms.
>
> We hope these revisions have fully addressed your concerns. If any points remain unclear or require further clarification, we would be very happy to provide additional details.
>
> Thank you again for your time and thoughtful feedback.

---

> ### Author Response · Authors · 2025-11-27
>
> Dear Reviewer wj9T,
>
> Thank you for your thoughtful comments and careful review. We have addressed each point in our rebuttal.
>
> If any part is still unclear, we’re glad to elaborate. We would be grateful if you could re-evaluate the paper considering these revisions.

---

> ### Comment · Reviewer_wj9T · 2025-11-28
>
> Thank you for the rebuttal.
>
> Please include the OmniContext benchmark, as well as the GEdit-Bench and EMUEdit results, in the final revision.
>
> Regarding the improvement to the GPT-based evaluation—"instead of evaluating all turns in one pass, we call API once for each turn, so that each API call focuses on one turn with previous turns as context."—I think this makes more sense now.
>
> Fig 14. show zero-shot generalization ability on general editing tasks, which seems impressive.
>
> I think the authors have addressed most of my concerns with extensive experimentation, and I will increase my score accordingly.
>
> I have one follow-up question regarding the model’s generalization ability on reference-based in-context editing. Specifically, I am curious whether the model can still perform in-context editing without being trained on the OmniGen dataset. For Figure 20, is this ability achieved only after the SFT stage? Could the authors provide some insight into this?

---

> ### Author Response · Authors · 2025-11-28
>
> **Thank you very much for your thoughtful comments and for letting us know that you intend to raise the score**. We truly appreciate the time you devoted to reviewing our work, and your positive reassessment of the benchmarks, automatic evaluation, and zero-shot generalization ability.
>
> We will include the OmniContext benchmark, as well as the GEdit-Bench and EMUEdit results, in the final revision.
>
> Regarding your question on in-context editing, we had provided the experimental setting in Sec. E.2 Appendix of the previous manuscript: the results in Fig. 20 were obtained using our model **fine-tuned on X2I2** [m].
>
> To clarify this further, we have added a more detailed explanation in the caption of Fig. 20 in the newly revised version (highlighted in magenta):
>
> > Our model is fine-tuned on the X2I2 dataset, but **the shown transferred concepts (\eg, background, color, and expression) are uncommon in X2I2**. It demonstrates the **strong generalization ability** conferred by pre-training on video-based interleaved data.
>
> [m] OmniGen2: Exploration to Advanced Multimodal Generation. arXiv'25

---

### Meta-Review · Area_Chair_7Z2g · 2026-01-06

**Summary:**

This paper presents VINCIE, a framework for in-context image editing that learns directly from video data without relying on curated image-editing pairs. It converts videos into interleaved multimodal sequences and trains a diffusion transformer using three proxy tasks: next-image prediction, current segmentation prediction, and next-segmentation prediction. The authors also introduce MSE-Bench, a new multi-turn editing benchmark. VINCIE demonstrates strong performance on multi-turn editing tasks and shows promising generalization to single-turn editing, composition, and story generation. Reviewers praised the novel use of video data, the well-designed pipeline, and the introduction of MSE-Bench. However, they raised concerns about the narrow definition of in-context editing, over-reliance on GPT-4o for evaluation, the need for comparisons on more recent benchmarks (e.g., GEdit-Bench, EMUEdit), the clarity of the method's novelty, and potential limitations in handling subtle edits or camera movements.

**Reviewer Concerns:**

In the rebuttal, the authors effectively addressed the majority of reviewers' concerns. They clarified the definition of in-context editing and provided new examples to demonstrate broader capabilities like reference-based composition and story generation. They supplemented the evaluation by adding performance comparisons on the GEdit-Bench, EMUEdit, and OmniContext benchmarks. Significant improvements were made to the GPT-based evaluation protocol by implementing per-turn assessments with finer-grained metrics (Prompt Following, Semantic Consistency). The authors also provided additional analyses on segmentation strategies, context usage, and the model's ability to generalize to creative edits, which were supported by new qualitative examples. Furthermore, they clarified the novelty of their approach, distinguishing it from prior video-based methods by emphasizing the interleaved session data paradigm. However, one underlying concern remains only partially mitigated: the fundamental reliance on existing VLMs for evaluation and data annotation. While the authors defended the use of GPT-4o by showing alignment with human judgment and consistency across other VLMs, and while they implemented filtering strategies to mitigate annotation errors, the inherent limitations of current VLMs in performing nuanced, multi-step judgments and capturing fine-grained visual differences persist as a potential weakness in the overall pipeline's robustness.

**Reviewer Scores:**

The authors' detailed rebuttal effectively addresses the core concerns raised in the initial reviews, paving the way for universally positive final evaluations. For reviewers concerned with evaluation breadth and the definition of "in-context," the authors supplemented results with three additional benchmarks (GEdit-Bench, EMUEdit, OmniContext) and provided qualitative evidence of broader capabilities. To mitigate issues regarding the reliability of GPT-based assessment, they implemented a refined, per-turn evaluation protocol with finer-grained metrics and demonstrated consistent trends across multiple AI judges. Questions about novelty and technical implementation were countered by clarifying the paradigm's focus on interleaved session data, explaining the distinct roles of segmentation tasks, and providing thorough ablation studies on context usage and inference strategies. While the fundamental reliance on external VLMs for annotation and evaluation remains a noted methodological dependency, the comprehensive experimental additions and clarifications provided substantively strengthen the paper's contributions and empirical validation, which would justify each reviewer raising their score to a clear acceptance threshold.

---

### Decision · Program_Chairs · 2026-01-26

Accept (Poster)